

COMPUTO

ISSN 2824-7795

# Detecting adverse high-order drug interactions from individual case safety reports using computational statistics on disproportionality measures

Jules Bangard ⬤[1]    Institut de Recherche Mathématique Avancée, UMR 7501 Université de Strasbourg et CNRS 7 rue René-Descartes, 67000 Strasbourg, France

Einar Holsbø ⬤    Faculty of Science and Technology, UiT-The Arctic University of Norway, PO, Box 6050 Stakkevollan, N-9037 Tromsø, Norway

Kristian Svendsen ⬤    Faculty of Health Sciences, UiT the Arctic University of Norway, Tromsø, Norway

Vittorio Perduca ⬤    CNRS, MAP5, Université Paris Cité, F-75006 Paris, France

Étienne Birmelé ⬤    Institut de Recherche Mathématique Avancée, UMR 7501 Université de Strasbourg et CNRS 7 rue René-Descartes, 67000 Strasbourg, France

Date published: 2026-01-15    Last modified: 2026-01-15

**Abstract**

Adverse drug interactions are a critical concern in pharmacovigilance, particularly as the controlled environment of clinical trials often lacks the scale and diversity to detect rare events or drug interactions. While spontaneous reporting systems (SRS) provide the necessary breadth for post-market surveillance, identifying these interactions within such large-scale data remains a significant computational challenge. This study introduces a computational framework for adverse drug interaction detection, leveraging disproportionality analysis on individual case safety reports. By integrating the Anatomical Therapeutic Chemical classification, the framework extends beyond drug interactions to capture hierarchical pharmacological relationships. This enables exploration of the space of drug interactions beyond pairwise interactions. To address biases inherent in existing disproportionality measures, we employ a hypergeometric risk metric, while a Markov Chain Monte Carlo algorithm provides robust empirical p-value estimation for the risk associated to cocktails. A genetic algorithm further facilitates efficient identification of high-risk drug cocktails. Validation on synthetic and FDA Adverse Event Reporting System data demonstrates the method's efficacy in detecting established drugs and drug interactions associated with myopathy-related adverse events. Implemented as an R package, this framework offers a reproducible, scalable tool for post-market drug safety surveillance.

*Keywords:* Disproportionality Analysis, Drug-Interactions, Genetic Algorithm, Markov Chain Monte Carlo, Pharmacovigilance, Spontaneous Reporting Systems

# Contents

---

[1]Corresponding author: jules.bangard@math.unistra.fr

# 1   Introduction

There are inherent limitations of clinical trials (RCTs) done before a medication is authorised in terms of cost, surveillance time, size and a lack of diversity of patients included (e.g. that pregnant woman, children and elderly often are not included) (Sanson-Fisher et al. 2007). Another issue is that RCTs are limited in their ability to detect rare adverse drug reactions (ADRs) and potential drug interactions (DDIs) since they usually only include a few thousand participants at most. Moreover, the use of multiple medications is often an exclusion criterion in RCTs, leaving gaps in the assessment of polypharmacy risks. As a result, tested drug combinations tend to be limited to those where physicians and pharmacologists have existing clinical experience or hypotheses about potential interactions (Heijden et al. 2002). This means that monitoring of the safety of medications and the

potential for DDIs after marketing is essential. This monitoring is often referred as post-authorization pharmacovigilance.

Elderly individuals are commonly prescribed multiple medications due to the increasing prevalence of comorbidities associated with aging. One study suggested that patients aged 75 or older in Austria took on average 7.5 drugs (Schuler et al. 2008). Among this demographic, 10% of hospitalizations were attributed to ADRs, and 18.7% of these ADRs were caused by a combination of more than one drug, a drug-drug interaction. More broadly, a meta-analysis attributed approximately 22.2% of hospitalizations caused by adverse drug reactions to DDIs (Dechanont et al. 2014).

Disproportionality analysis (DA) encompass widely used methods in pharmacovigilance to detect ADRs by assessing the frequency of adverse events (AE) associated with specific drug consumption relative to what would be expected. Most established disproportionality methods like the proportional reporting ratio (PRR)(Evans, Waller, and Davis 2001), reporting odds ratio (ROR)(Eugene P. van Puijenbroek et al. 2002), Bayesian Confidence Propagation Neural Network (BCPNN) (Bate et al. 1998), and the Gamma Poisson Shrinker (GPS) (DuMouchel 1999) allow for the identification of signals that warrant further investigation in the context of single drug assessment. The tree-based scan statistic Heo et al. (2024) takes advantage of the tree structure of the Anatomical Therapeutic Clinical (ATC) drug classification in order to allow to select a drug family rather than a single drug, a family corresponding to a subtree of the ATC tree. Likelihood-ratios tests then allow to choose the most relevant family.

In order not to overinterpret high proportions in case of rare drug cocktails, it is possible to derive a score from an independence test on the contingency table counting the presence or absence of the adverse event depending on the consumption of the drug cocktail. (Gosho et al. 2017) use a chi-square statistic to do so, while (Ahmed et al. 2010) rather use the Fisher exact test which has the advantage of being non-asymptotic.

Fewer DAs methods are available to use for multiple drugs consumption, such as the $\Omega$ shrinkage method (Norén et al. 2008). This method allows the use of a DA measure in order to detect sets of the type *Drug-Drug-Adverse Event* and stands as the standard measure when it comes to detecting the interactions of two drugs. Adaptation of the PRR for a single drug has been proposed like the Concomitant Signal Score (CSS) (Noguchi et al. 2020) and the PRR adaptation for DDIs (Wang et al. 2020).

Other methods than DA measures exist in order to detect DDIs. Among them, well known methods encompass logistic models (Eugène P. van Puijenbroek et al. 2000; Van Puijenbroek et al. 1999) and association rules methods (Noguchi et al. 2018; Ibrahim et al. 2016). One can refer to multiple reviews for a more detailed overview of existing methods (Ibrahim et al. 2021; Hauben 2023).

Computational pharmacovigilance methods are typically applied to Spontaneous Reporting Systems (SRS), which aggregate Individual Case Safety Reports (ICSRs)—real-world data submitted by healthcare professionals, manufacturers, and patients. Such reports contain informations about intake of medications for each patients as well as their experienced AE. While SRS databases, such as the FDA Adverse Event Reporting System (FAERS), provide the scale necessary to detect rare events, they possess inherent limitations. These include under-reporting, where only a fraction of adverse events are captured (Bate and Evans 2009), and various reporting biases, such as the notoriety bias (Pariente et al. 2007). Crucially, the spontaneity of these reports often results in an incomplete recording of all drugs taken by a patient. Furthermore, because the exact timing of drug administration is frequently omitted, it is difficult to distinguish between simultaneous intake and medications taken at different times within the same reporting window. This temporal ambiguity potentially limits the precision of interaction analyses, as the observed cocktail may represent a sequence of exposures rather than a concurrent pharmacological event. Consequently, because the total number of patients exposed

to a drug is unknown, these databases cannot provide true incidence rates, necessitating the use of disproportionality analysis to identify safety signals (Almenoff et al. 2005).

While previous work showed that further practical experiences should be of interest for high-order drug interaction testing (Tekin et al. 2018), it is hard to know which exact combination one should test and easy to miss an important combination in the overwhelming space of high-order drug cocktails.The present article proposes a novel method to do so. It requires the choice of a scoring function able to measure, for a given AE, the importance of the disproportionality for any medication set, including single medications. It then takes advantage of the tree structure of the ATC classification to explore the space of medication sets in two ways: a genetic algorithm looking for the sets of high score, and a MCMC algorithm learning the distribution of the scores for a given cocktail size in order to evaluate how extreme a high score is. The approach is also original with respect to the fact that a medication denotes either a drug or a drug family in the sense of an internal node of the ATC classification.

The developed algorithms, available in the corresponding R package **emcAdr**(Bangard 2025), aim to guide pharmacology researchers toward drug combinations that might correspond to drug interactions and require more rigorous monitoring.

## 2 Methods

### 2.1 ATC Tree and Cocktail Definition

Multiple classifications of drug active ingredients exist, one being the ATC classification. This system is structured hierarchically and can be represented as a tree with five levels containing, at the time, 6809 nodes. The leaves of this tree are the active ingredients, while the first level consists of nodes representing organs or systems affected by the descendant active ingredients. The remaining nodes correspond to therapeutic or pharmacological families.

By applying a Depth-First Search algorithm to enumerate each node of the ATC tree, we can define a drug cocktail as a set of integer corresponding to specific nodes in the tree. More formally, a cocktail $C$ of size $k \geq 1$ is defined by $C = (x_1, \ldots, x_k) \in \Delta^k$ where $k \in \mathbb{N}$, and $\Delta$ represents the set of numbered nodes of the ATC tree $T$. Figure 1 shows examples in a simplified tree, the green nodes denoting the considered cocktail. Note that sets of drugs of size one are also considered, and by abuse of langage are also refered to as cocktails.

Considered cocktails can include internal nodes of the tree (thus representing families of active ingredients), allowing for the detection of more general signals. For example, paracetamol might send a weak signal, while if we move up the tree, analgesics as a whole could represent a stronger and more general signal. Therefore, all patients taking at least one drug from this drug family will be considered in the computation of the risk. That notion is equivalent to the notion of cut in the tree-scan method (Kulldorff, Fang, and Walsh 2003).

### 2.2 Cocktail Risk Characterization

In the field of pharmacology, accurately characterizing the risk associated with drug administration is a complex task. The aim of the developed method is to search the space of cocktails to maximize a score indicating if individuals taking a given drug combination have a higher risk of an AE. While the algorithm is designed to be flexible and can be run with any scoring function, we focus on a comparison of established metrics to justify our selection (see Section 3.1).

### 2.2.1 Scoring Functions in Pharmacovigilance

To evaluate the disproportionality of drug cocktails presence for a particular AE, several statistical measures have been proposed in the literature:

**Proportional Reporting Ratio (RR)** A standard measure defined as the ratio of the probability of an AE occurring in the group exposed to the cocktail versus the non-exposed group. Its application in signal generation from spontaneous reports was notably discussed by (Evans, Waller, and Davis 2001).

**PRR for Drug-Drug Interactions (PRR)** Unlike the single-drug PRR, this adaptation for drug combinations evaluates whether a combination represents a synergistic risk. Specifically, as discussed by (Wang et al. 2020), a signal is often defined by comparing the lower bound of the confidence interval of the combination's risk to the maximum risk observed for the individual drug components.

$\Omega$ **Shrinkage** A Bayesian measure specifically developed for drug-drug-event combinations. As proposed by (Norén et al. 2008), it utilizes a "shrinkage" approach to reduce false positive signals in cases with limited data by pulling the observed association toward the value expected under independence.

**Concomitant Signal Score (CSS)** Introduced as an improved detection criterion, this metric aims to enhance the detection of DDI signals by building upon the PRR framework, as detailed by (Noguchi et al. 2020).

**Proposed Score: Hypergeometric Disproportionality** While the methodology can accommodate various scores, we propose the use of a score derived from the Fisher exact test. As highlighted by (Ahmed et al. 2010), this approach has the advantage of being non-asymptotic. Consider a dataset of $N$ patients, among which $K$ experience the adverse event AE. Let $n_C$ be the number of people taking a cocktail $C$ and $x$ the number of patients taking $C$ and experiencing AE. We define the risk to be:

$$H(C) = -\log(\mathbb{P}(X \geq x))$$

where $X \sim \mathscr{H}(n_C, K, N)$ follows a hypergeometric distribution. This measure effectively functions as the log p-value under the null hypothesis that the number of people with AE in a uniform sample of $n_C$ out of $N$ people follow the hypergeometric distribution. It has the distinct advantage of accounting for the total number of patients exposed to the cocktail; for instance, $H(C)$ will assign a higher risk to a cocktail taken by 100 patients with 50 AE cases than to a cocktail taken by 10 patients with 5 AE cases, whereas simple ratios like the PRR might treat them identically. Such hypergeometric measures are well-established in other high-dimensional fields like bioinformatics for functional enrichment analysis(Grossmann et al. 2007).

## 2.3 Distinguishing Combined Risk from True Interaction

While the $H(C)$ score effectively identifies cocktails associated with a significant increase in the AE frequency, it does not inherently distinguish a synergistic pharmacological interaction from situations corresponding to the independent addition of individual effects or an effect driven by a subset of the cocktail (e.g., "innocent" medications co-prescribed with a high-risk drug).

To address this, we propose a post-treatment of detected signals using Firth's penalized likelihood logistic regression (Firth 1993). This method is particularly suited for spontaneous reporting systems as it reduces bias in maximum likelihood estimates and provides a solution to the problem of separation, which often occurs with rare adverse events. Using the *logistf* R package(Heinze, Ploner, and Dunkler 2023), we estimate a model for a given cocktail $C$ (e.g., $d_1, d_2$) as follows:

$$\text{logit}(P(AE = 1)) = \beta_0 + \beta_1 \mathbb{1}_{d_1} + \beta_2 \mathbb{1}_{d_2} + \beta_3 \mathbb{1}_{d_1 \cap d_2}$$

An interaction is confirmed if the interaction coefficient $\beta_3$ is positive and statistically significant. This indicates that the risk associated with the co-prescription exceeds the additive effect of the individual components on the logit scale. While the emcAdr package allows the use of Firth's penalized regression to handle the rare events and separation issues. Users may implement the post-treatment of their choice.

## 2.4 Disproportionality Identification

### 2.4.1 High-risk Drug Cocktails Identification

The number of possible cocktails is $2^L$, where $L$ denotes the number of nodes in the ATC tree, and even for a given cocktail size $k$, the number of possibilities is still $\binom{L}{k}$. It is therefore not possible to compute $H(C)$ for each possible cocktail $C$ included in the dataset. Instead, to explore the space of cocktails and locate those at high-risk of AE, we use a genetic algorithm (Pétrowski and Ben-Hamida 2017). It simulates the evolution of a population of cocktails according to the principle of natural selection, in order to search for the most performing individuals with respect to an evaluation criterion based on $H$.

The steps are the following, the algorithm repeating steps 2 and 3 until a user-defined number of iterations is reached.

**Initialization.** The genetic algorithm's population consists of $m$ cocktails. These cocktails are randomly initialized and can vary in size.

**Evaluation and selection.** At iteration $n$, the current population $P_n$ undergoes an evaluation and selection phase. The evaluation computes the score $H(C)$ for each cocktail $C$ in the population. A new population $Q_{n+1}$ of the same size is drawn by sampling $m$ times a pair of cocktails in the original population and copying the one with highest score. Note that this step performs selection as the expectation of the number of copies in $Q_{n+1}$ of a cocktail from $P_n$ is an increasing function of its score. A penalization is however applied to avoid a uniformisation of the population, as further explained in Section 2.4.2.

**Stochastic modification.** Mimicking the genetic drift in nature, stochastic modifications occur in the population in order to explore the large space of cocktails. First, a crossover operation allows two cocktails to exchange information. Here, the crossover involves exchanging sub-trees between two cocktails as follows:

- an internal node $v$ of the ATC tree is randomly selected;

- the nodes of the subtree rooted at $v$ are exchanged between the two sequences.

After performing the crossover, a mutation is applied to the resulting individuals, chosen from two types. The first type is a local mutation which changes a randomly selected node of the cocktail to one of its free neighboring nodes. This mutation is further explained in Section 2.5. The second type is an addition/deletion mutation which operates as follows, with $k$ being the cocktail length and $\alpha$ a chosen hyperparameter:

- with probability $\min(1, \frac{\alpha}{k})$, a node uniformly drawn from $\Delta$ is added to the sequence;

- with probability $1 - \min(1, \frac{\alpha}{k})$, a uniformly drawn node from the cocktail is removed.

An example of crossover and mutations can be seen on Figure 1 (a), (c) and (d).

214   Applying those modifications to $Q_{n+1}$ yields a population $P_{n+1}$ which is used to loop at step 2.

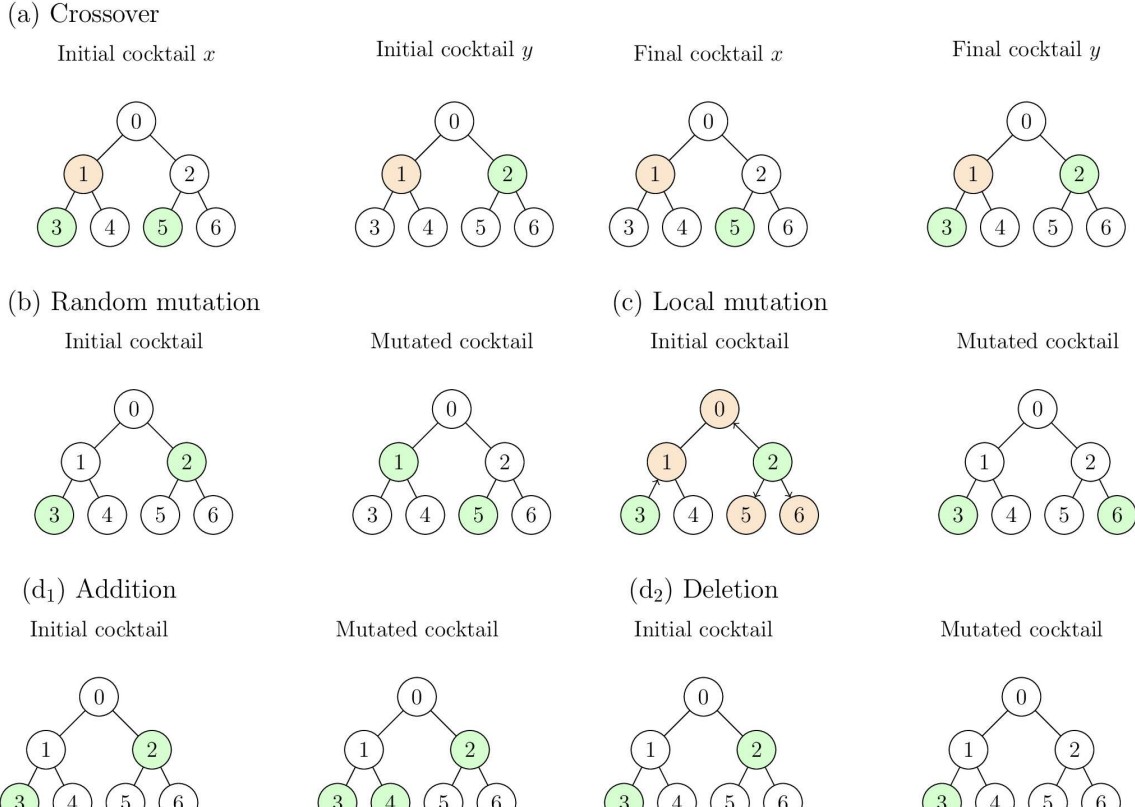

Figure 1: Cocktail modifications used for the genetic algorithm (a, c and d) and the MCMC algorithm (b and c). Green nodes are part of considered cocktails. In Crossover, the orange node represents the selected internal node whose subtrees are being swapped. In local mutations, orange nodes represent legal moves

### 2.4.2   Distance between drug cocktails and cocktail penalisation

216   If at some point of the algorithm, an individual of $P_n$ has a high score compared to other individuals,
217   the genetic algorithm may converge to a population consisting entirely of that cocktail. To avoid
218   this uniformisation phenomenon, similar cocktails are penalized in the evaluation phase as follows,

$$H_{pen}(C) = \frac{H(C)}{\sum_{C_i \in \mathscr{C}} \text{Sim}(C, C_i)}$$

219   The computation of the similarity $\text{Sim}(C, C')$ is based on a distance inspired by the Levenshtein
220   distance (Levenshtein et al. 1966). However, unlike the traditional Levenshtein distance, sequences
221   are treated as unordered sets. For two drug cocktails, $C_1$ and $C_2$, of sizes $n_1$ and $n_2$, the distance
222   $d(C_1, C_2)$ is defined as the minimal cost required to transform $C_1$ into $C_2$ using three elementary
223   operations.

224   • $\text{Ins}_a(C)$ consists of adding $a$ to the cocktail $C$.
225   • $\text{Del}_a(C)$ consists of deleting $a$ from the cocktail $C$.
226   • $\text{Sub}_{a,b}(C)$ consists of substituting $a \in C$ by $b$.

227   An associated cost is defined for each operation.

**Substitution.** The cost associated with the substitution operation is chosen to be consistent with the conceptual similarity of cocktails. If $a$ is a drug belonging to $C$, the cost should increase as the drug $b$ diverges further from drug $a$. For example, if $a$ is a drug, and $b$ a drug family that contains $a$, the cost should be moderate. Conversely, if $b$ is a drug family not containing $a$ the cost should be higher. This distance is thus defined by the maximal distance between $a$ and $b$ to their Lowest Common Ancestor.

**Insertion, Deletion.** The deletion and insertion cost are chosen as $\frac{depth(T)}{2}$. This choice implies that a substitution always costs less than a deletion followed by an insertion. The latter are used only when the two cocktails do not have the same length.

A transformation $f$ from $C_1$ to $C_2$ is a composition of elementary operations that go from $C_1$ to $C_2$. The associated cost $\mathrm{cost}(f)$ is defined as the sum of the cost of the operations used in $f$. Finally, $d(C_1, C_2) = \min_f \mathrm{cost}(f \,:\, f(C_1) = C_2)$.

Finally, the maximum distance between $C_1$ and $C_2$ being $(n_1 + n_2)\frac{depth(T)}{2}$, we define the similarity as

$$Sim(C_1, C_2) = 1 - \frac{2D(C_1, C_2)}{(n_1 + n_2)depth(T)}$$

The computation of the similarity is achievable in $\mathcal{O}(n_1 \times n_2 \times \mathrm{depth}(T) + |\Delta|)$ operations in the worst case. The algorithm to compute the similarity is detailed in the Section 5.1 (see Algorithm 1).

### 2.4.3 Output clustering

Despite the diversity mechanisms integrated into the genetic algorithm, not all drug cocktails retrieved within the genetic algorithm population are unique. It is moreover common to encounter solutions that are merely variations of others, differing only by transformations such as changing a node in the ATC tree to its parent or child. To streamline analysis and enhance the efficiency, a post-treatment clustering of similar drug cocktails is implemented. This method allows to focus on the most risky cocktails within each cluster or to interpret pharmaceutically clusters rather than individual cocktails.

To do so, drug cocktails are embedded into a two-dimensional space using the UMAP algorithm (McInnes, Healy, and Melville 2018), which aims to preserve similarity in the latent space. This representation enables the effective use of conventional machine learning clustering algorithms in $\mathbb{R}^2$. Specifically, the DBSCAN algorithm (Ester et al. 1996) is applied to identify clusters of similar drug cocktails with an intuitive way of choosing hyperparameters.

## 2.5 Approximate P-Value Assignment for Drug Cocktails

Once a list of cocktails of high score has been established by the genetic algorithm, an important step of the analysis is to assign them p-values to decide if they are significant.

Consider a cocktail $C$ of size $k$ in the list and its score $S_{obs}$. Denote by $H_0$ the null hypothesis according to which a cocktail does not favor the AE, and by $N_0$ the number of cocktails of size $k$ for which $H_0$ is the truth. Similarly, $H_1$ represents the alternative hypothesis of a cocktial favoring the AE and $N_1$ is the number of such cocktails of size $k$. The p-value corresponding to $S_{obs}$ is then $\mathbb{P}_{H_0}(S > S_{obs})$, but cannot be estimated without modelisation hypothesis under $H_0$.

However, if $\mathbb{P}_M(S)$ refers to the marginal distribution, that is the score distribution for the cocktails both in $H_0$ and $H_1$,

$$\mathbb{P}_M(S > S_{obs}) = \mathbb{P}_{H_0}(S > S_{obs})\frac{N_0}{N_0 + N_1} + \mathbb{P}_{H_1}(S > S_{obs})\frac{N_1}{N_0 + N_1}$$

so that

$$\frac{N_0 + N_1}{N_0}\mathbb{P}_M(S > S_{obs}) - \frac{N_1}{N_0}\mathbb{P}_{H_1}(S > S_{obs}) \leq \mathbb{P}_{H_0}(S > S_{obs}) \leq \frac{N_0 + N_1}{N_0}\mathbb{P}_M(S > S_{obs})$$

Under the reasonable assumption $N_1 \ll N_0$, the probability $\mathbb{P}_M(S > S_{obs})$ on the marginal distribution can thus be used as an approximation of the p-value. Note that the upper bound yields that, under the weak hypothesis that less than 10% of the cocktails of size $k$ favor the AE, the real p-value is at most equal to 1.11 times its approximation. Moreover, the approximated p-value can be estimated by computing the score of cocktails sampled in the general cocktail population. The proposed method therefore focuses on a sampling scheme in the general population and uses the approximated p-value to declare significance.

A naive sampling of cocktails considers almost only cocktails taken by no patient in the dataset. A Metropolis-Hastings MCMC algorithm is thus considered, as it can be used by conditioning on the fact that all visited cocktails are present in the dataset (Au and Beck 2001).

To employ such an algorithm, it is necessary to define a state space $\mathcal{C} = \{C_1, ..., C_p\}$, a computable target measure $f(C_i)$, and conditional laws $q(.|C_i)$ under which simulation is possible and new states can be proposed.

**State set.** The state set is made of all cocktails of $k$ drugs for a fixed $k$.

**Proposal law.** The proposal law is defined as a mixture of two mutation laws of the current cocktail. They operate as follows:

- Random mutation consists of a completely random movement in the cocktail space.

- Local mutation involves a local movement relative to the structure of the drug tree. Here, a node $x_p$ of the state $C_i$ is changed to one of its free neighboring nodes.

At each iteration, the random and local mutations have probability $p_R$ and $1 - p_R$, where $p_R$ is a hyperparameter.

Figure 1 (b, c) presents examples of a random and a local mutations.

**State evaluation.** The evaluation of a drug cocktail is based on the score $H(C)$. The chosen target measure is then:

$$f_T(C_i) = \frac{1}{Z(T)} \times e^{\frac{H(C_i)}{T}}$$

where $Z(T) = \sum_C e^{\frac{H(C)}{T}}$. $T$ is a parameter known as temperature, which modulates space exploration by more readily accepting cocktails with moderate scores (high $T$) or, conversely, by strongly favoring combinations of drugs with high scores (low $T$).

The acceptance probability of cocktail $C_{i+1}$ from cocktail $C_i$ is given by:

$$\min\left(1, \frac{f_T(C_{i+1})}{f_T(C_i)} \times \frac{q(C_i|C_{i+1})}{q(C_{i+1}|C_i)}\right)$$

The theory related to the Metropolis-Hastings algorithm (Robert and Casella 2004) ensures that the empirical distribution of $f_T(C_i)$ for the constructed cocktail chain converges to the distribution

of $f_T(C)$. A very long realization of such a walk, therefore, allows for the approximation of the distribution that can be used to determine an empirical p-value for the score of a cocktail of interest. This enable the possibility to say whether or not a high-risk is truly significant (e.g. among the top 5% of scores). It defines what a high risk is in our case.

## 2.6 Datasets

### 2.6.1 Simulated data

Multiple datasets were simulated to evaluate the method performance against known outcomes. The datasets, designed to challenge the algorithm, simulate various patient scenarios. Each patient record includes prescribed medications and the corresponding occurrence of an adverse event $AE$.

The first dataset is composed of 200,000 patients, and has the following characteristics:

- 1% of the patients take a size-3 drug cocktail $C_1$ and have a $\frac{1}{100}$ chance of having $AE$.
- 1% take a size-3 drug cocktail $C_2$ and have a $\frac{1}{200}$ chance of having $AE$.
- 1% take a size-2 drug cocktail $C_3$ and have a $\frac{1}{100}$ chance of having $AE$.
- 1% take a size-2 drug cocktail $C_4$ and have a $\frac{1}{200}$ chance of having $AE$.

A small percentage of the dataset (1.5% per combination) are combinations of 2 out of the 3 drugs from $C_1$ and $C_2$, but with no risk of $AE$. This helps to mitigate the false identification of sub-cocktails of $C_1$ and $C_2$ as high risk cocktail because those who take two drugs of these cocktails will almost surely take the remaining drug of the cocktail.

The remaining 87% of the datasets consists of patients assigned with random cocktails drawn uniformly. The size $s$ of each cocktail is drawn according to a Poisson distribution with $\lambda = 4$ (mean size of drugs cocktail taken by patients in the dataset). For each cocktail, $s$ nodes of the ATC tree are selected uniformly, with each combination assigned an adverse event with probability $\frac{1}{15000}$.

Three others datasets were similarly constructed, the difference lying in the size of the cocktails inducing $AE$. One has only size-two cocktails, other only size-three cocktails and the last one, size-two, three and four cocktails.

### 2.6.2 FAERS Data

As a proof-of-concept on real data, the method was also assessed using the FAERS dataset, which consists of ICSRs submitted by healthcare professionals, consumers, and manufacturers. These reports include details on patient drug intake and the side effects experienced. The methods were deployed our on FAERS data from the second quarter of 2013 to the second quarter of 2015, the restriction to a two-year window corresponding to the will of limiting computational time.

Significant refinement of the FAERS data was required. Duplicate reports were removed, retaining only the report with the most recent ID. Subsequently, a link was established between the prescribed drugs and the ATC codes of each active ingredient. This process involved matching drug names to their respective active ingredients and converting these ingredients to their corresponding ATC codes. The DiANA dictionary (Fusaroli et al. 2024) facilitated the standardization of FAERS drug names to ATC codes. It is important to note that the DiAna dictionary handles combination products by splitting them into their constituent active substances rather than assigning a single ATC combination code. Reports with unmatchable drug names in the DiANA dictionary were excluded, reducing the dataset from 2,043,231 to 1,612,931 patients.

For this study, we focused on myopathy as the selected adverse event outcome. It is a clinically concerning condition with a sufficient number of reported cases in the dataset (536 cases). To validate

our results, we compared the identified drug-myopathy associations with known drugs already established to cause myopathy Valiyil and Christopher-Stine (2010). Code for data refinement is available on GitHub.

# 3 Results

## 3.1 Score Comparison

To support the use of the hypergeometric score $H(C)$ introduced in Section 2.2, we compared it to other risk scores on simulated data. The simulated dataset comprises pairwise drug combinations. A subset of these cocktails was generated under the alternative hypothesis $H_1$, meaning they are specifically associated with an increased probability of the AE (see Section 2.6.1). The remaining cocktails were sampled under the null hypothesis $H_0$, representing a baseline risk without specific interaction. Figure 2 illustrates the performance of the risk scoring methods for detecting high-risk drug combinations introduced in Section 2.2.1. Each subplot displays the score values for cocktails representing true solutions, sampled under $H_1$ (green) and cocktails not representing true solutions, sampled under $H_0$ (red). The bottom-right panel presents the Precision-Recall (PR) curve, comparing the detection power of the scores in identifying high-risk cocktails. PRR is a special score because its value is either one or zero, which allows the computation of only one value of precision and recall. *PRR* is therefore represented by a single point rather than a PR curve.

PR curves of the different scoring methods are mainly comparable, confirming the conclusions of (Candore et al. 2015). However, as illustrated by the jitter plots, the hypergeometric score and the $\Omega$ Shrinkage measure better rank and isolate true solutions from other cocktails. It must be emphasized that the $\Omega$ Shrinkage measure is difficult to compare using a threshold, as the original article (Norén et al. 2008) suggests signaling a cocktail when the score exceeds zero. However, in the simulations, the computed score never exceeded zero, as depicted by the x-axis of its jitter plot. Despite these considerations, these two methods are the least biased toward cocktails taken by only a few patients.

The good behavior of the hypergeometric score and its easy generalisation to cocktails of any size justify the use of that scoring function for the exploration of the cocktail space.

## 3.2 Application to the Simulated Dataset

### 3.2.1 Estimation of Risk Distribution

Risk distribution was estimated for size-two drug cocktails on Section 2.6.1 dataset. Estimation of risk distribution for higher cocktails sizes are possible but it is nearly impossible to compare it to the true distribution as it is computationally prohibitive to obtain. The distribution estimated by the MCMC algorithm, is compared to the true risk distributions in Figure 3.

The left panels display the distributions of risk scores for both the estimated (top-left) and true (bottom-left) risk values. Both distributions share a similar shape, with the majority of risk scores concentrated at low values, as 95% of the scores fall below 11. However, some differences can be observed, particularly in the tail of the distribution, where cocktails of higher risks are under-represented in the estimated distribution.

The right panel of Figure 3 presents a Probability-Probability (PP) plot (top) and a Quantile-Quantile (QQ) plot (bottom), comparing the quantiles and probabilities of the estimated and true risk distributions. While the right panel of the figure demonstrates good agreement at lower risks, where most of the data lie, deviations at higher risk values suggest that the estimated distribution slightly underrepresents the risk for more extreme values.

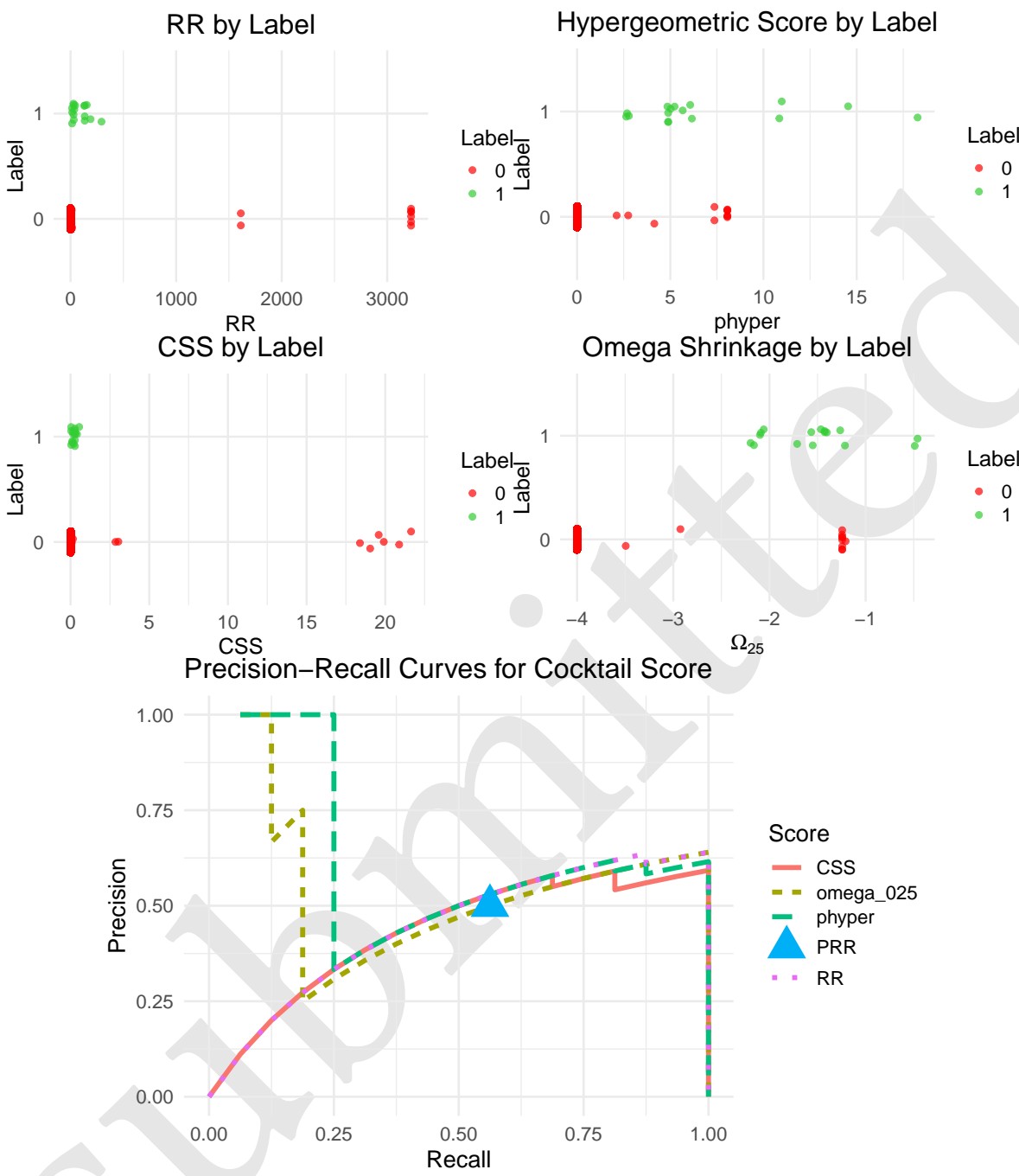

Figure 2: Comparison of scores for cocktail of size-two. Each dots denotes a cocktail while his risk computed on the formerly presented synthetic dataset (Section 2.6.1) is showed on the x-axis. Red dots represent cocktails that do not induce adverse event (negative controls), green ones represent cocktail inducing adverse event (positive controls). The bottom-right corner shows the Precision-Recall curves for each score. The perfect classification corresponds to the upper right corner. The areas under the curves allow to compare different methods. RR, PRR, Omega shrinkage, phyper and the CSS are defined in Section 2.2.1.

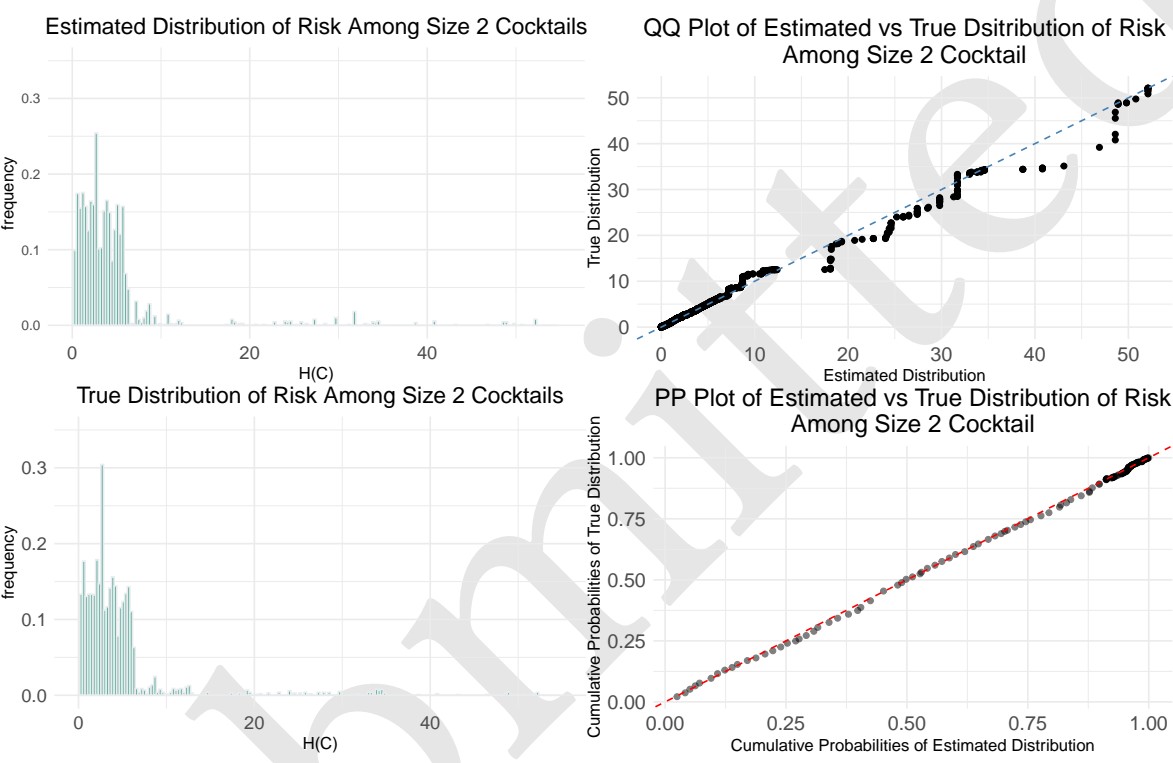

Figure 3: Comparison of estimated and true risk distributions for size-two drug cocktails. Left panels show comparison of risk distribution among size-two cocktails, right panels allows to compare probabilities and quantiles of both distributions

Results indicate that the method performs well in estimating risk scores for the majority of cocktails, capturing the overall risk distribution with reasonable accuracy. The slight underestimations which are present for high-risk cocktails are not a problem since the interest of the method is to assign p-values. P-values are still reliable as shown in the PP-plot, Figure 3. Consequently, we can assign robust empirical p-values to any size-two cocktails based on their risks. Furthermore the calibration under the null hypothesis for the p-values attributed by the algorithm is assessed in Section 5.2.

### 3.2.2   Genetic algorithm output and clustering

The genetic algorithm was applied to the simulated dataset to identify high-risk drug cocktails. Multiple runs of the algorithm were conducted using different hyperparameter sets (population size, number of generations, parameter $\alpha$) to ensure the visit of different regions of the space of possible cocktails. The results were subsequently concatenated to create a comprehensive list of high-risk cocktails.

The genetic algorithm successfully identified nearly all high-risk size-two and size-three drug cocktails in the simulated dataset (Figure 4). For size-two cocktails, the algorithm consistently found the exact high-risk combinations. However, for size-three cocktails, the algorithm sometimes identified cocktails that were very close to the true high-risk combinations, missing only one drug from the correct cocktail in a few cases (oftentimes, choosing parent nodes instead of the actual drugs).

To streamline the analysis of the large set of results, significant cocktails were filtered using the empirical p-value by setting a threshold of 5%. Clustering techniques were then applied to group similar cocktails together. As discussed in Section 2.4.3, the UMAP algorithm has been used for dimensionality reduction, followed by the DBSCAN clustering method. This post-processing step allows to reduce redundancy by grouping cocktails that differed only slightly, such as by substituting a drug for another within the same pharmacological family. Such cocktail would have similar medical interpretations.

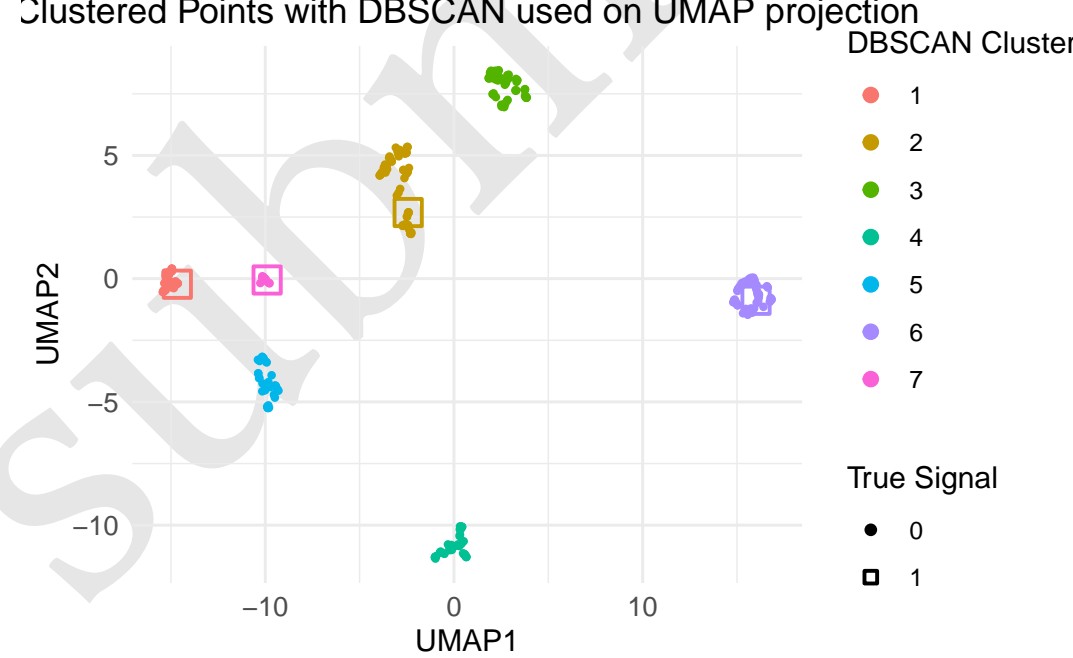

Figure 4: Clustering of high-risk drug cocktails identified by the genetic algorithm on the simulated dataset. Each dot represents a drug cocktail, true cocktail solutions are the centers of the squares

Figure 4 illustrates the results of the clustering process after mapping the selected cocktails in

the plane. Each point represents a drug cocktail, and the 7 automatically determined clusters are represented by the colors.

The first conclusion is that the clusters are well separated in terms of the editing distance defined in Section 2.4.2. The genetic algorithm thus identified seven regions of the cocktails space of high score that are different in terms of interpretation as their cocktails are far away between classes. On the other hand, cocktails in the same clusters are close, indicating a possible pharmacological interpretation of the cluster.

The second conclusion is that the algorithm effectively found the true solutions or at least very similar solutions. Indeed, the embedding of the true solutions yields the four squares, that clearly belong to four different clusters among the seven. The method therefore identifies and separates regions of the cocktail space containing the true solutions. That simulation thus enhances the confidence that investigating the high-score cocktails returned by the methods may allow to detect relevant phenomena.

## 3.3 Application to the FAERS Spontaneous Reporting Data

### 3.3.1 Estimation of Risk Distribution

The risk estimation method was applied to the FAERS spontaneous reporting dataset presented in Section 2.6.2. Figure 5 presents a comparison between the estimated risk distribution and the true risk distribution for size-two drug cocktails. The left panel of the figure shows the distributions of risk scores, while the right panel presents the QQ plot and the PP plot, comparing the quantiles and probabilities of both distributions.

The histogram reveals that the estimated distribution aligns well with the true distribution for the majority of cocktails with lower risk scores. However, deviations begin to emerge in the tail of the distribution. Specifically, 15 of the riskiest cocktails in the true distribution were not captured by the estimated distribution as we see in the QQ plot. This explains the observed shift in the QQ plot at the highest quantiles since higher risk cocktails have not been found by the MCMC algorithm, highlighting the need of a complementary method like the genetic algorithm in order to find riskiest cocktails.

Despite this slight deviation, the empirical p-values remain robust for both lower and higher-risk cocktails as shown by the PP-plot in the Figure 5.

### 3.3.2 Genetic algorithm output and clustering

The genetic algorithm was applied on the FAERS data focusing on the myopathy AE by running 180 parallel executions of the genetic algorithm on varying population sizes (from 100 to 1000 cocktails per generation). The whole procedure run in less than 8 hours on a 24-core server.

Cocktail sizes present in the merged final populations vary from 1 to 6. The MCMC algorithm was run for each cocktail size in this range to assign an empirical p-value to each solution. All solutions having a p-value lower than 0.05 were kept. No multiple testing correction was made in order to avoid false negatives, that is disregarding interesting cocktails, even if this may inflate the number of false positives. 682 drug cocktails composes the final list. Applying Benjamini-Hochberg corrections to control the FDR at level 5% and 10% respectively kept the first 107 and 564 elements of the list.

The clustering procedure was applied to the 682 cocktails, leading to 15 clusters. To validate the results, the first 150 solutions were tagged with the families they correspond to, if any, without knowledge of the clustering's result. To do so, some drugs or drug families known to be linked to myopathy adverse events were considered, that is hypolipemic drugs, Colchicine, corticosteroids,

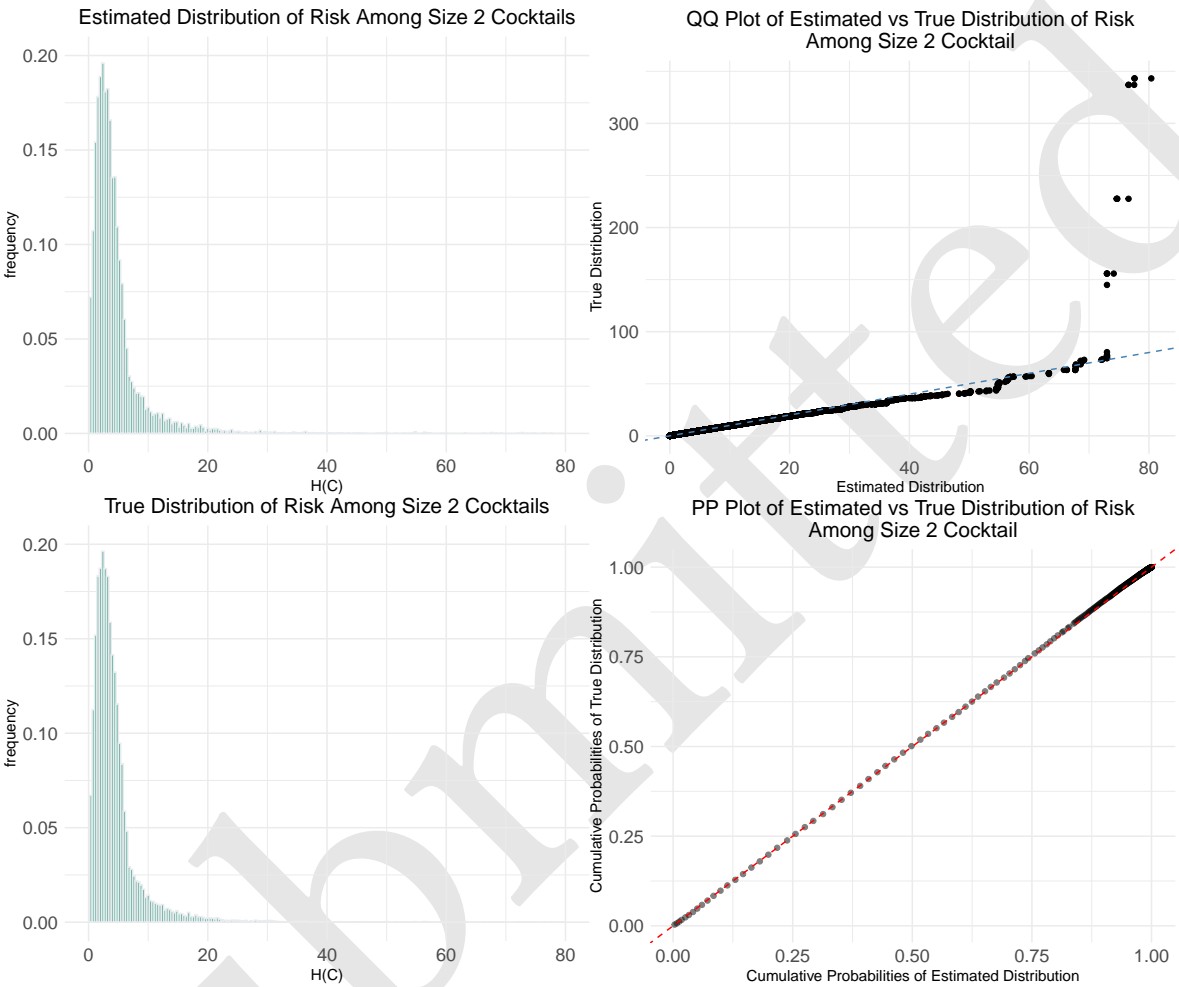

Figure 5: Comparison of estimated and true risk distributions for size-two drug cocktails on the FAERS dataset presented Section 2.6. Left panels show comparison of risk distribution among size-two cocktails, right panels allows to compare probabilities and quantiles of both distributions. Left distributions are truncated to 80 for visualization purposes. A few cocktails have a risk around 300 as shown on the Y-axis of the QQ plot.

Ciclosporine, Beta blocking agents, Fluoroquinolones, and anti-malaria drugs (Miernik, Matusiewicz, and Olesińska 2024; Hall, Finnoff, and Smith 2011; Valiyil and Christopher-Stine 2010).

The final result is a table of 682 rows, one per selected cocktail, indicating its composition, the number of patients taking it, the number of patient taking it and facing myopathy, hypergeometric and RR score. It also contains the cluster it belongs to and the tagged families. The entire table is available on this link. Table 1 summarizes the cluster assignments for the top 150 cocktails identified in the genetic algorithm. Among these, eleven cocktails could not be assigned to any drug or drug family listed in the table headers. Note that a cocktail can be associated with more than one drug or drug family.

Table 1: Summary of clustering applied to the identified solutions. Each row corresponds to a cluster. The 150 cocktails with the highest risk were analyzed. Box $(i, j)$ in the table represents the number of cocktails in cluster $i$ that include the drug or drug family $j$

| Cluster | Hy-polipemic Drugs | Colchicine | Steroids | Cy-closporine | Beta blocking agents | Domperi-done | Fluoro-quinolones |
|---|---|---|---|---|---|---|---|
| 1 | **31** | 0 | 3 | 0 | 0 | 0 | 4 |
| 2 | 0 | **11** | 0 | 4 | 0 | 0 | 0 |
| 3 | 7 | 0 | 0 | 0 | **19** | 0 | 0 |
| 4 | 0 | 0 | 0 | 0 | 0 | **50** | 0 |
| 5 | 0 | 0 | 0 | 0 | 0 | **19** | 0 |
| 6 | 0 | 0 | 0 | 0 | 0 | 0 | **4** |

This table shows the method's capability to detect known ADRs from the FAERS dataset and highlights the ability of the clustering method to group cocktails with similar pharmacological interpretations. Specifically, cluster 1 corresponds to cocktails containing Hypolipemic Drugs, cluster 2 to Colchicine and Cyclosporine, cluster 3 to beta blocking agents and cluster 6 to Fluoroquinolones. These clusters align with previously reported pharmacological associations (Miernik, Matusiewicz, and Olesińska 2024). Similarly, clusters 4 and 5 correspond to Domperidone-containing cocktails which have been associated with adverse effects, including potential cardiac complications, as reported by the British Medicines and Healthcare products Regulatory Agency (Medicines and Healthcare products Regulatory Agency 2014).

Interestingly, additional drug families reported in (Miernik, Matusiewicz, and Olesińska 2024), such as anti-malarial drugs, were also identified but at later ranks. For instance, anti-malarial drugs are primarily grouped in cluster 9, as shown in the complete solution table.

# 4 Conclusion

As co-medication becomes increasingly common, there is a growing need for methods capable of detecting signals of harmful drug combinations from the available large databases for further assessment. The proposed method addresses this need by identifying signals and assigning them a p-value using a hypergeometric disproportionality analysis measure. Additionally, the method enables the identification of broader signals within the ATC hierarchy by proposing "cocktails" of not only active substances but also chemical, therapeutic, and anatomical families, leveraging the hierarchical classification of active substances.

Application on synthetic datasets demonstrated that using the hypergeometric score reduces the false positives from cocktails taken by a small number of patients, enhancing the robustness of

the measure. The results on these datasets of our MCMC algorithm to estimate the distributions of cocktail risks were encouraging, as the estimated distributions closely aligned with the true distributions, indicating reliable p-value assignment. Furthermore, the genetic algorithm effectively identified the majority of the harmful cocktails, with a high success rate, highlighting its efficiency in navigating the large solution space.

Applying this method to previous FAERS data for the myopathy adverse event yielded promising results. A literature review confirmed the intersection between the identified signals and drugs known to have a higher likelihood of causing myopathy, demonstrating the effectiveness of the proposed methodology. Results also indicate that certain drug combinations are more strongly associated with myopathy than individual medications. Notably, the cyclosporine/colchicine combination exhibits a higher hypergeometric score (73.1 vs. 63.4) and PRR (879.4 vs. 57.1) compared to colchicine alone, reinforcing the importance of analyzing drug interactions in pharmacovigilance. This combination is known to be more likely to induce myopathy (Ducloux et al. 1997).

Furthermore, our approach identified a size-four drug cocktail (metformin, prasugrel, bisoprolol, simvastatin) associated with an increased risk signal (hypergeometric score = 72.2, RR = 3060.6). This combination was observed in nine patients, all of whom experienced the adverse event. While this finding demonstrates the feasibility of detecting higher-order drug interactions, we emphasize that no clinical validation is currently available for this specific combination. This underscores both the potential of the method in identifying complex drug interactions and the need for further validation through complementary studies.

The application of the post-treatment logistic regression to the strongest signals provided further clinical refinement. For instance, the interaction between cyclosporine and colchicine was confirmed as an additional interaction effect: even after accounting for individual drug effects, the coefficient associated with the combined cocktail remained positive and highly significant ($p = 1.02e^{-6}$). This result corroborates existing medical literature indicating a potentiation of muscle toxicity.

Conversely, the analysis of the identified size-four cocktail (metformin, prasugrel, bisoprolol, simvastatin) revealed that the risk signal was almost entirely explained by a size-three sub-cocktail (metformin, bisoprolol, simvastatin). In this instance, the addition of the fourth medication did not yield a significant additional interaction effect. This finding highlights the importance of the post-treatment step in identifying the "core" of an interaction.

This approach can also be extended to other settings where adverse events are explored. For instance, the method could be applied using the ICD diagnosis classification or the MedDRA system, both of which are hierarchical classifications. Such an application would facilitate the identification of symptoms associated with the consumption of drug combinations.

The proposed method is implemented as an R package *emcAdr*, available on GitHub and on the CRAN (Bangard 2025). A tool allowing the process of quarterly FAERS xml files to csv file directly usable by our method is also available on [Github](Github).

For researchers and stakeholders it is crucial to remember that our method is hypothesis generating as are other signal detection methods in single medications. These methods aim to generate as few false negative results as possible but with the trade-off of more false positives. This means that any results should firstly be assessed for biologically plausibility as well as validated in a more formal causal framework such as RCTs and target trial emulations. Furthermore, while spontaneous reporting systems have a key role in pharmacovigilance, they have known drawbacks as underlined in the introduction. They however provide important knowledge about the safety of drugs, and with our method also drug cocktails.

## 525 Funding

This work was partially supported by the "PHC AURORA" programme (project number: 49704QC), funded by the French Ministry for Europe and Foreign Affairs, the French Ministry for Higher Education and Research and the Norwegian Council for Research.

## 5 Appendices

### 5.1 Appendix A : Distance Pseudo-code Algorithm

---
**Algorithm 1** CalculateDistance
---
**Require:** Matrix $M$, Lists $idxC_1$, $idxC_2$
**Ensure:** Cost of the distance calculation
1: $height \leftarrow 0$
2: $cost \leftarrow 0$
3: **while** $idxC_1 \neq \varnothing$ **and** $idxC_2 \neq \varnothing$ **do**
4:     **for all** $iC_1 \in idxC_1$ **do**
5:         **for all** $iC_2 \in idxC_2$ **do**
6:             **if** $M[iC_1][height] = M[iC_2][height]$ **then**
7:                 $cost \leftarrow cost + height - \min\Big(M[iC_1].count(M[iC_1][0]), M[iC_2].count(M[iC_2][0])\Big) + 1$
8:                 **remove** $iC_1$ **from** $idxC_1$
9:                 **remove** $iC_2$ **from** $idxC_2$
10:                 **break**
11:             **end if**
12:         **end for**
13:     **end for**
14:     $height \leftarrow height + 1$
15: **end while**
16: $insertion\_cost \leftarrow \text{ATC\_HEIGHT}/2$
17: $cost \leftarrow cost + (|idxC_1| + |idxC_2|) \times insertion\_cost$
18: **return** $cost$

---

Algorithm 1 computes the distance between two cocktails, $C_1$ and $C_2$, based on three inputs.

The first input is an integer matrix $M$. This matrix represents the nodes of $C_1$ and $C_2$, as well as their corresponding parent nodes. The matrix has dimensions $(PN, \text{ATC\_HEIGHT})$, where $P$ and $N$ denote the sizes of $C_1$ and $C_2$, respectively. For example, consider the simplified tree shown in Figure 1 of the article. In this tree, $C_1$ could be the cocktail containing only the node $\begin{bmatrix} 3 \end{bmatrix}$, and $C_2$ could be the cocktail containing only the node $\begin{bmatrix} 2 \end{bmatrix}$ ($P = 1$ and $N = 1$).

The corresponding matrix $M$ would be:

$$M = \begin{bmatrix} 3 & 1 & 0 \\ 2 & 2 & 0 \end{bmatrix}$$

The second input consists of $idxC_1$ and $idxC_2$, which represent the indices of the rows in $M$ that contain the nodes of $C_1$ and $C_2$, respectively.

## 5.2 Appendix B : Calibration Under the Null

To assess the validity of the approximation of the p-value, we plotted the empirical distribution of p-values for cocktails presumed not to increase the adverse-event rate on the synthetic dataset. The theoretical p-values following a Uniform(0, 1) distribution, the approximated ones should exhibit a distribution close to uniform if the approximation is valid. Figure 6 shows a histogram of these p-values, which appears approximately flat.

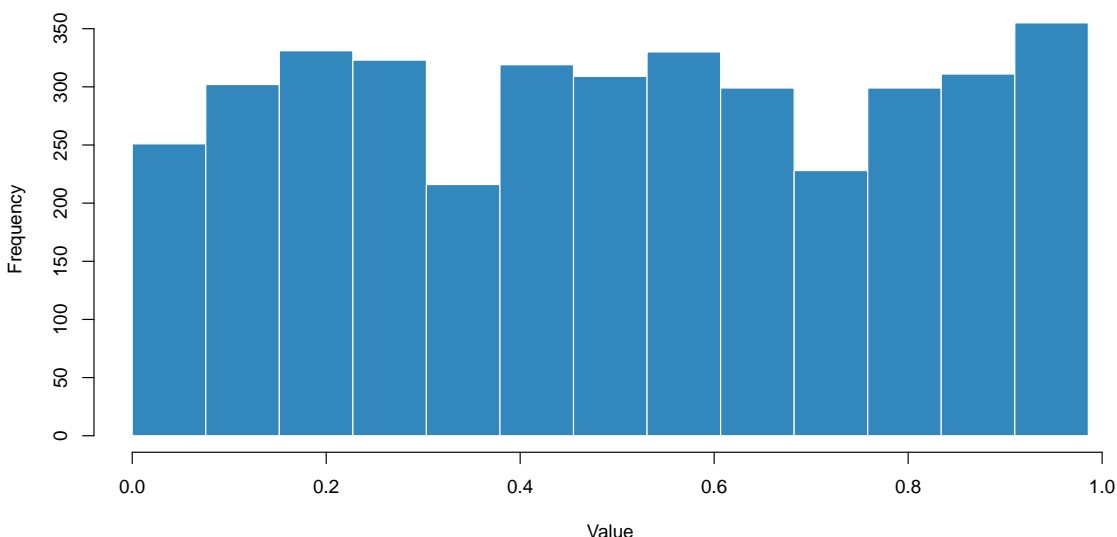

Figure 6: Histogram of p-values for cocktails presumed to satisfy the null hypothesis on synthetic dataset (no elevated adverse-event risk)

## 5.3 Appendix C : FAERS Clustering

Figure 7 shows the clustering applied to the results of the genetic algorithm ran on the FAERS datasets.

## 5.4 Appendix D : Algorithmic Complexity

The total time complexity of the high-risk cocktail identification is given by:

$$\mathcal{O}(G \cdot P \cdot T_{eval})$$

Where $G$ represents the number of generations (or MCMC iterations) and $P$ represents the population size of the GA (this equals one for the MCMC algorithm). The evaluation time for a single cocktail, $T_{eval}$, is defined by the matching logic required to count occurrences across the database:

$$T_{eval} = \mathcal{O}(N \cdot K \cdot R)$$

- $N$ (Number of Reports): The algorithm iterates once through the database to compute the hypergeometric score for a given cocktail. The runtime scales linearly with the number of patients.

- $K$ (Cocktail Size): The number of drugs in the candidate cocktail. In this study, $K$ is typically small since there is a limit of drug a patient take.

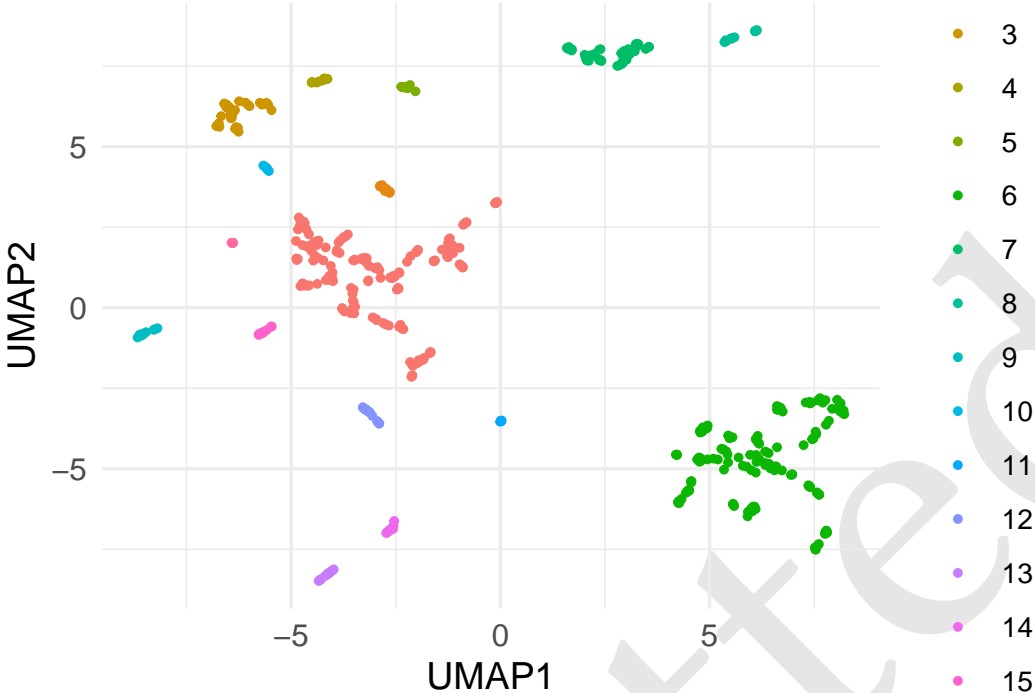

Figure 7: Clustering of High-Risk Drug Cocktails Identified by the Genetic Algorithm on the FAERS dataset. This Figure allows to see proximity between multiple clusters and better understand why some cluster exhibits similarities.

- $R$ (Drugs per Report): The number of medications listed in a single patient report. On our data the mean size of drug cocktails per patient is approximately 2.2. Because $K$ and $R$ are bounded, the complexity simplifies to $O(N)$ per candidate evaluation.

# 6   Code and Results

See the complete results of application of methodology on the FAERS dataset on this link.

Code to reproduce the experiments conducted in this article can be found here :

- GitHub link of the data refinement code : JulesBa-Git/FAERS-xml-parser.
- GitHub link to the latest release of the R package of the complete methodology : JulesBa-Git/emcAdr.
- The code of the methodology is also available on the CRAN.
- The code to obtain figures is available on the figures/scripts folder of this repository.

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

# Session information

```
R version 4.5.1 (2025-06-13)
Platform: aarch64-apple-darwin20
Running under: macOS Tahoe 26.1

Matrix products: default
BLAS:   /Library/Frameworks/R.framework/Versions/4.5-arm64/Resources/lib/libRblas.0.dylib
LAPACK: /Library/Frameworks/R.framework/Versions/4.5-arm64/Resources/lib/libRlapack.dylib;  LAPACK

locale:
[1] fr_FR/UTF-8/fr_FR/C/fr_FR/fr_FR

time zone: Europe/Oslo
tzcode source: internal

attached base packages:
[1] stats     graphics  grDevices datasets  utils     methods   base

other attached packages:
[1] stringr_1.6.0 umap_0.2.10.0 ggplot2_3.5.2 dbscan_1.2.2  dplyr_1.1.4
[6] emcAdr_1.2

loaded via a namespace (and not attached):
 [1] Matrix_1.7-3      gtable_0.3.6      jsonlite_2.0.0    compiler_4.5.1
 [5] renv_1.1.5        tinytex_0.57      tidyselect_1.2.1  Rcpp_1.1.0
 [9] png_0.1-8         scales_1.4.0      yaml_2.3.10       fastmap_1.2.0
[13] reticulate_1.43.0 lattice_0.22-7    R6_2.6.1          labeling_0.4.3
[17] generics_0.1.4    knitr_1.50        tibble_3.3.0      openssl_2.3.3
[21] pillar_1.11.0     RColorBrewer_1.1-3 rlang_1.1.6      stringi_1.8.7
[25] xfun_0.52         cli_3.6.5         withr_3.0.2       magrittr_2.0.3
[29] digest_0.6.37     grid_4.5.1        askpass_1.2.1     lifecycle_1.0.4
[33] vctrs_0.6.5       RSpectra_0.16-2   evaluate_1.0.4    glue_1.8.0
[37] farver_2.1.2      rmarkdown_2.29    tools_4.5.1       pkgconfig_2.0.3
[41] htmltools_0.5.8.1
```

