# OpenReview forum: "Detecting adverse high-order drug interactions from individual case safety reports using computational statistics on disproportionality measures"
_Computo — Accepted by Computo_

### Review · Reviewer_7TJo · 2025-11-24

**Summary Of Contributions:**

In this paper, the authors propose a new method for detecting drug-drug interactions using spontaneous reporting data from pharmacovigilance. Notably, the proposed method adapts a genetic algorithm to explore the vast set of possible drug-drug combinations.

**Audience:**

Yes

**Claims And Evidence:**

No

**Requested Changes:**

Below are some more specific comments (major and minor):

Abstract:

I don’t understand the first sentence, as it seems to imply that spontaneous reports are not suited to detecting drug interactions, yet your application is based on spontaneous reports.

Introduction:

I think the introduction should present the type of data used (i.e. spontaneous reports), explain what they are and how they are collected, and discuss their inherent limitations.

Methods:

I think the methods section should include a description of other existing methods, particularly those that will be compared in the manuscript. For example, it is unclear whether the authors used the three PRR criteria (N > 3, PRR statistics > 2, and chi statistics > 4). For their toy example, I believe there are some differences.

The statistic proposed by the authors is  based on the hypergeometric distribution, and I believe they are very similar to those proposed by Ahmed et al. (Biometrics, 2010; Clin Pharmacol Ther, 2010), who proposed the Fisher exact test.

I don't understand how the statistics measure an interaction effect. My understanding is that the effect of a potential interaction should be greater than the individual effects of each drug. Here, it seems that the authors are measuring the effect of several drugs in combination without making any reference to the effects of individual drugs. In cases where two drugs are co-prescribed, but only one is responsible for the adverse effect, the H(C) statistic could be high, even in the absence of interaction.

2.3.1 The sketch of the algorithm is unclear to me: the algorithm has three steps. However, the authors then mention a 'reproduction' phase. What is this? In the evaluation and selection phase, I don't understand how cocktails are selected.

Modification and population replacement: it is not clear from the description how the algorithm can improve from one iteration to another.

2.3.2: 'When a signal is identified': At this stage in the manuscript, it is unclear what the authors mean by 'signal'. In my understanding, a signal implies some kind of threshold. But what is the detection rule?
The sizes of C1 and C2 are referred to as either n1/n2 or n and m.

2.4: If I understand correctly, the aim of the algorithm is to obtain the null distribution of H(C). However, it seems to me that the output of the algorithm is the marginal distribution of H(C), which contains a mixture of true and false DDIs. How do the authors derive a p-value from such a distribution?

Figure 3: There are problems in the legend at the top of the plot.

2.5.1: 1% seems a fairly high combination for a size 2 or 3 cluster. How did the authors choose these proportions?

2.5.2: ICSRs should be specified.

2.5.2: Why is the analysis restricted to two years (2014–2015), when FAERS data are available from 2012 up to 2025?

The authors base their detection on a 5% p-value threshold without accounting for multiple testing.

Results:
3.1 The methods compared to H(C) should be described briefly. At this stage, we have no idea what PRR, CSS and omega shrinkage are (they were referred to as the Greek letter in the introduction, by the way). I think it is strange to describe the simulation comparison metrics in the results section. Additionally,  on which simulated data are based these results (Figure 2 only indicate that the results are based on size 2 cocktails) ?

3.2.2: The description of the results is very vague. The authors explain that they combined the results by varying the parameters and using post-processing steps. Ultimately, the performance of the approach is difficult to evaluate. Furthermore, the results concerning cocktails of sizes 3 and 4 are barely discussed.

3.3 Figure 5. I have the same question as in 2.4. The authors state that their genetic algorithm failed to correctly capture the tail of the marginal distribution of H(C), as illustrated in Figure 5. If I understand correctly, this is precisely because the tail corresponds to potential true signals that are very rare. However, this is a marginal distribution, not the distribution of the statistic under the null hypothesis, so I wonder how the authors can derive proper p-values from it.

Figure 5 lower left : is truncated since the x axis should go from 0 to more than 300. At least, this should be indicated in the caption.

Figure 5 : Titles of each sub plot should be reformated as some of them are truncated

L339 I wonder why the authors were interested in studying drug coktails of size 1 since they do not correspond to drug drug interactions.

If I understand correctly, the method is evaluated in its ability to identify cocktails containing at least one drug belonging to 6 drug families known to cause myopathy. II don’t understand how it validate the ability of the method to detect drug drug interactions.In particular a cocktail can contain a drug responsible for the adverse events and other co-prescribed drugs which are innocent bystanders.

**Strengths And Weaknesses:**

Overall, I am afraid I find the article lacking in clarity in many areas, such as for instance:

(i) the application context;

(ii) the description of the methods with which their approach is being compared; and

(iii) some aspects of the proposed method.

Furthermore, I am unsure whether the proposed statistic is appropriate for studying drug-drug interactions. Finally, I find that both the simulation study and the application struggle to account for the performance of the methods.

---

> ### Author Response · Authors · 2026-01-16
> **First answers to specific remarks of reviewer 7TJo**
>
> **Abstract: I don’t understand the first sentence, as it seems to imply that spontaneous reports are not suited to detecting drug interactions, yet your application is based on spontaneous reports.**
>
> We rephrased the first sentence of the abstract.
>
> **Introduction: I think the introduction should present the type of data used (i.e. spontaneous reports), explain what they are and how they are collected, and discuss their inherent limitations.**
>
> Introduction has been revised in order to give an overview of what contains a spontaneous reports as well as what are its inherent limitations.
>
> We moreover developed the end of the introduction to better expose the motivations of the paper.
>
> **Methods: I think the methods section should include a description of other existing methods, particularly those that will be compared in the manuscript. For example, it is unclear whether the authors used the three PRR criteria ($N > 3$, PRR statistics $> 2$, and chi statistics $> 4$). For their toy example, I believe there are some differences.**
>
> From a general perspective of the methods description, we added a short definition of the scoring functions we compare. We moreover emphasize that it is necessary to select a score to continue the procedure, but that the methodology of the genetic and MCMC algorithms are not score dependent. This part was mainly rewritten.
>
> Concerning PRR, we found the three mentioned criteria at page 6 of the guideline document of the European Medicines Agency available here
> (https://www.ema.europa.eu/en/documents/regulatory-procedural-guideline/draft-guideline-use-statistical-signal-detection-methods-eudravigilance-data-analysis-system_en.pdf).
>
> It however indicates that one can either use a threshold on the PRR value and $\chi^2$ statistic, or a lower bound on the $95\%$ confidence interval of the PRR (it is written in the case of a single drug, but it is stated page 5 that the calculations can be extended to drug-drug interactions). The definition and decision method we used for the PRR in our simulated study are the ones of Wang et al. (2020), which are based on the comparison of lower bounds of confidence intervals.  They therefore seems  fair to us in terms of method comparison. Moreover, we'd prefer not to set emphasis on that discussion in the paper, as this criterion anyway does to our knowledge not generalize to cocktails of any size, which is needed for our algorithm.
>
> **The statistic proposed by the authors is based on the hypergeometric distribution, and I believe they are very similar to those proposed by Ahmed et al. (Biometrics, 2010; Clin Pharmacol Ther, 2010), who proposed the Fisher exact test.**
>
> We thank the reviewer for pointing those references we missed. Hypergeometric score and Fisher exact test are indeed equivalent. We added the citations and modified the text accordingly.
>
> **I don't understand how the statistics measure an interaction effect. My understanding is that the effect of a potential interaction should be greater than the individual effects of each drug. Here, it seems that the authors are measuring the effect of several drugs in combination without making any reference to the effects of individual drugs. In cases where two drugs are co-prescribed, but only one is responsible for the adverse effect, the H(C) statistic could be high, even in the absence of interaction.**
>
> The reviewer is correct that disproportionality measures like $H(C)$ do not inherently measure synergistic effects. We have clarified that the $H(C)$ score is a discovery tool used to navigate the vast combinatorial space.
>
> We think it is interesting to point out cocktails with high disproportionality even if their effect correspond to the addition of the individual drugs, as the sum of small effects can lead to a risk that is considered too high. The presence of synergistic effects can however be tested in a post-treatment phase using a penalized logistic model. We added a paragraph proposing to do so using a logistic regression including individual drug indicators and an interaction term. It allows to test if the combination's effect is significantly greater than the sum of its parts on the logit scale. This possibility has been added to the package, but to preserve the flexibility of the latter, the function is separate from the main algorithms and the user may apply other post-treatment methods if desired.

---

> ### Author Response · Authors · 2026-01-16
> **Second answers to specific remarks of reviewer 7TJo**
>
> **2.3.1 The sketch of the algorithm is unclear to me: the algorithm has three steps. However, the authors then mention a 'reproduction' phase. What is this? In the evaluation and selection phase, I don't understand how cocktails are selected.**
>
> **Modification and population replacement: it is not clear from the description how the algorithm can improve from one iteration to another.**
>
> We modified quite substantially the description of the genetic algorithm. The term reproduction was referring to the modification and population replacement, which was indeed confusing. We hope it is clearer now.
>
> **2.3.2: 'When a signal is identified': At this stage in the manuscript, it is unclear what the authors mean by 'signal'. In my understanding, a signal implies some kind of threshold. But what is the detection rule? The sizes of C1 and C2 are referred to as either n1/n2 or n and m.**
>
> This phrasing was unclear. What we meant by signal was a region of the space to explore where there are locally high scores. What we observe if we do not apply the penalization of 2.3.2 is that the whole population converges to very similar cocktails (one of the colors in the simulation data). We rephrased that sentence.
>
> We replaced $n$ and $m$ by $n_1$ and $n_2$.
>
> **2.4: If I understand correctly, the aim of the algorithm is to obtain the null distribution of H(C). However, it seems to me that the output of the algorithm is the marginal distribution of H(C), which contains a mixture of true and false DDIs. How do the authors derive a p-value from such a distribution?**
>
> The reviewer is right, we are estimating the distribution of $H(C)$ on the population of all cocktails, including those under the null and the alternative distribution. The number of cocktails under $H_1$ is very low compared to the number of possible cocktails, and we therefore approximate the theoretical p-value by the estimation under the marginal. It should however be clarified. We modified the text in that direction and added the bounds of the p-value in terms of the probability on the marginal law.
>
> **Figure 3: There are problems in the legend at the top of the plot.**
>
> Legend of Figure 3 has been modified
>
> **2.5.1: $1\%$ seems a fairly high combination for a size 2 or 3 cluster. How did the authors choose these proportions?**
>
> The goal of the simulated dataset was not to be as realistic as possible, but to have a toy example for the proof of concept of the following facts: 1) the hypergeometric score better discriminates the real cocktails among the methods extending to any-size cocktails;  2) the MCMC algorithm achieves a good approximation of the true $H(C)$ distribution on size two cocktails for which the exact distribution is tractable, validating our approximate p-values 3) the genetic algorithm selection contains solutions that are close to the perfect ones.
>
> The different proportions were chosen in terms of expected number of simulated patients having the ADR: around $20$ of them have it due to respectively $C_1$ and $C_3$, $10$ of them due to $C_2$ and $C_4$, and $13.3$ totally randomly. The total expected number of positive patients is of $73.3$ and the cocktails to be found are therefore responsible for an ADR among $3.5$ or $7$ respectively.
>
> Those choices are perhaps to optimistic but seemed reasonable to check that the method is able to find the right cocktails in that case.
>
> **2.5.2: ICSRs should be specified.**
>
> More details have been added to Introduction and section 2.5.2.
>
> **2.5.2: Why is the analysis restricted to two years (2014–2015), when FAERS data are available from 2012 up to 2025?**
>
> The reason of this choice is again because the article is methodological and not a medical investigation of the causes of myopathy. As a proof of concept, the application to two years of data seemed quite enough due to the volume of data. Pre-treatment of the data is not immediate (Note: the corresponding code is available), and running our algorithms on the considered data required 8 hours on a local cluster. An exhaustive treatment would have required much more computational cost without gaining insight in the methodology. We thus decided not to run it, but it is perfectly possible using the available code. Further details about complexity are presented in Appendix D.
>
> **The authors base their detection on a 5% p-value threshold without accounting for multiple testing.**
>
> As written in the original article, we kept all solutions with a p-value less than $0.05$ to avoid false negatives, that is disregarding interesting cocktails, even if this may inflate the number of false positives. We however computed the corrected p-values with FDR rates of $5\%$ and $10\%$, which kept respectively the $107$ and $564$ first values of the table. Those results have been added in the text.

---

> ### Author Response · Authors · 2026-01-16
> **Third answers to specific remarks of reviewer 7TJo**
>
> **Results: 3.1 The methods compared to H(C) should be described briefly. At this stage, we have no idea what PRR, CSS and omega shrinkage are (they were referred to as the Greek letter in the introduction, by the way). I think it is strange to describe the simulation comparison metrics in the results section. Additionally, on which simulated data are based these results (Figure 2 only indicate that the results are based on size 2 cocktails) ?**
>
> We have added brief definitions for PRR, RR, Omega Shrinkage, and CSS in Section 2.2. This ensures that the reader can understand the conceptual differences between these metrics before evaluating their performance on our simulated datasets.
>
> Indeed, the description of the dataset on which the results of 3.1 were computed lacked clarity. We rephrased the paragraph and added information about data on which these score has been computed. We hope the text is clearer now.
>
> **3.2.2: The description of the results is very vague. The authors explain that they combined the results by varying the parameters and using post-processing steps. Ultimately, the performance of the approach is difficult to evaluate. Furthermore, the results concerning cocktails of sizes 3 and 4 are barely discussed.**
>
> We are not sure to completely understand the concerns of the reviewer for this point.
>
> The genetic algorithm is random by nature, and there is no guarantee that a single run will discover all the regions of the space containing high score solutions. We therefore run it several times, with different starting points and hyperparameters (size $m$ of the population, parameter $\alpha$ in the mutations, number of generations in the genetic algorithm) in order to explore different regions of the space. The post-processing step only consists in a concatenation of the best results of the different runs.
>
> Concerning the evaluation, the cocktails in that final list are clustered after a non-linear dimension reduction. It yields $7$ clusters, and if the $4$ true cocktails favoring ADR (which are of size $2$ or $3$) are also mapped in the low-dimensional space, we see that $4$ of the clusters correspond to cocktails equal or close to them. The size of the cocktails is here not the point, what matters is that the method, in the huge space of possibilities, returns for each true solution some cocktails that are close in terms of editing distance. We think it is good news in terms of the confidence we have that investigation by a pharmacologist of the high score cocktails returned by our method may allow to detect relevant phenomena.
>
> We rephrased partly this paragraph, hoping to make it clearer.
>
> **3.3 Figure 5. I have the same question as in 2.4. The authors state that their genetic algorithm failed to correctly capture the tail of the marginal distribution of H(C), as illustrated in Figure 5. If I understand correctly, this is precisely because the tail corresponds to potential true signals that are very rare. However, this is a marginal distribution, not the distribution of the statistic under the null hypothesis, so I wonder how the authors can derive proper p-values from it.**
>
> It is the MCMC algorithm that approximates the marginal distribution of $H(C)$. The genetic algorithm is designed to explore regions of the space where high scores are present, while the MCMC algorithm should give a general vision of the distribution of $H(C)$.
>
> The fact that the tail is not well approximated is indeed due to the fact that the MCMC random walk did not explore yet the whole space when it was interrupted, so that it indeed missed  some of the signals with extreme score.
>
> As mentioned in the answer of the previous comment, the notion of p-value has been replaced by the approximated p-value, as the marginal survival probability we estimate is close to the actual p-value. In particular, the upper bound added in 2.4 shows that it is possible to use the approximate p-value as an indicator of potential interest for a pharmacologist.
>
> **Figure 5 lower left : is truncated since the x axis should go from 0 to more than 300. At least, this should be indicated in the caption.**
>
> The x-axis in Figure 5 (left panels) was intentionally truncated to enhance the readability of the main distribution, as high-score outliers are sparse and barely visible on the histogram. We agree that this should be transparent; the figure caption has been modified to reflect this truncation. The QQ-plot remains the primary tool for comparing the distributions across the full range of values.
>
> **Figure 5 : Titles of each sub plot should be reformatted as some of them are truncated**
>
> We thanks the reviewer for this remark, titles have been reformatted.

---

> ### Author Response · Authors · 2026-01-16
> **Fourth and last answers to specific remarks of reviewer 7TJo**
>
> **L339 I wonder why the authors were interested in studying drug cocktails of size 1 since they do not correspond to drug drug interactions.**
>
> Our method allows to detect sets of medications that are significantly linked to the AE, whatever their size. The method was not especially designed to study cocktails of size $1$. However, excluding them would complexify the genetic algorithm (to forbid crossovers resulting into a single drug for instance), and would not yield any particular gain. Indeed, it is possible to exclude a posteriori sets of size $1$ from the results analysis (and the approximated p-values would not be modified as they are computed separately for each cocktail size).
>
> **If I understand correctly, the method is evaluated in its ability to identify cocktails containing at least one drug belonging to 6 drug families known to cause myopathy. I don’t understand how it validate the ability of the method to detect drug drug interactions.In particular a cocktail can contain a drug responsible for the adverse events and other co-prescribed drugs which are innocent bystanders.**
>
> The method was evaluated by selecting the cocktails of highest score, and conducting in parallel (and without interaction) the clustering step and the annotation in known families by a pharmacist (K. Svendsen). The good news are that our clusters correspond almost perfectly to those families, that is make sense and separate different mechanisms. That relevance in terms of pharmacological interpretation makes the proof of concept on FAERS data of interest.
>
> However, the reviewer is right in the sense that our method does list sets of cocktails that show a high disproportionality of people experiencing the AE, which may not correspond to an interaction. A post-processing step is thus needed to tackle that question, which is done  through a post-treatment using Firth’s penalized logistic regression (other post-treatments could be preferred by the user). This procedure was applied to some of the cocktails found on the FAERS dataset and a discussion on the results has been added. A mention was also added  to be careful not to interpret all found cocktails as interactions and be aware that harmless bystanders could be present.

---

### Review · Reviewer_YBby · 2025-12-22

**Summary Of Contributions:**

This paper introduces a computational framework for detecting potentially hazardous multi‑drug “cocktails” using spontaneous reporting system (SRS) data. The authors propose: (1) A probabilistic formulation in which the expected hazard of drug combinations is modeled using a hypergeometric distribution. (2) A heuristic search strategy based on a genetic algorithm to explore the combinatorial space of multi‑drug combinations and identify high‑risk candidates. (3) An unsupervised clustering step using DBSCAN to group similar high‑risk cocktails. (4) An MCMC‑based p‑value estimation procedure to approximate the significance of the observed hypergeometric hazard relative to simulated null distributions. (5) A simulation study designed to compare the behavior of the proposed risk measure against established disproportionality metrics (e.g., PRR, RR, Omega).

The methodology is applied to a subset of FAERS as a case study, illustrating how the proposed pipeline can highlight high‑risk multi‑drug combinations that may not emerge from traditional pairwise DDI analyses. The paper positions this integrated modelling, search, and clustering framework as a novel contribution to the computational analysis of drug–drug interactions in SRS data.

**Audience:**

Yes

**Broader Impact Concerns:**

Although the paper does not include a Broader Impact Statement, I believe adding one would be appropriate given the public‑health implications of the work. This study develops an algorithmic framework to identify potential drug–drug interactions using spontaneous reporting systems (SRS). Such methods, if misinterpreted or deployed without appropriate safeguards, may influence clinical decision-making, regulatory considerations, or public perception of drug safety.

I would therefore encourage the authors to briefly reflect on:
- The risk of misinterpretation of algorithmically identified associations, especially since SRS data suffer from underreporting, missing temporal information, and reporting biases.
- The possibility of false positives or false negatives, and how these may affect patient safety if results are communicated without appropriate caveats.
- The ethical responsibility to frame outputs as hypothesis‑generating rather than causal, particularly when using heuristic search algorithms (GA, DBSCAN) whose outputs are not statistically calibrated.
- The potential impact on stakeholders (clinicians, regulators, patients) if high‑risk drug combinations identified through the method are disseminated without appropriate contextualization.

A short paragraph addressing these points would strengthen the responsible‑research framing of the work.

**Claims And Evidence:**

No

**Requested Changes:**

Major suggestions

- *Limitations of spontaneous reporting systems (SRS).*
I recommend adding a discussion on the inherent limitations of SRS data regarding drug reporting. Due to the spontaneity of reporting, not all drugs taken are necessarily recorded, and the timing of exposures is not always specified. As a result, reported drugs may reflect simultaneous intake or different time windows, potentially limiting the accuracy of interaction analyses. Additionally, although FAERS allows reporters to specify interacting agents, this field is rarely populated. It may be informative to report how many cases in your FAERS subset include interacting substances.
- *Justification of methodological choices.*
1. *Choice of the hypergeometric distribution.*
Please provide a more detailed rationale for modelling the hazard of drug combinations using a hypergeometric distribution. Were alternative distributions considered? Why was the hypergeometric deemed most appropriate? Is there precedent in the literature, or was this decision model‑driven by the authors? Although rows 99–102 compare your measure to the PRR, other methods commonly applied to drug–drug interactions (DDIs) may offer relevant contrasts. For instance, the Omega method is based on a Gamma–Poisson model. What advantages does the hypergeometric approach provide relative to these well‑established alternatives?
2. *Choice of a genetic algorithm (GA) as a search method.*
To my knowledge, genetic algorithms are rarely used in pharmacovigilance, and more broadly, heuristic search algorithms have not been widely applied in spontaneous reporting systems. I strongly recommend adding a justification for preferring a GA over established SRS search or screening approaches, such as subset‑based methods (https://dl.acm.org/doi/10.1145/2110363.2110395) or tree‑based scan statistics (https://doi.org/10.1038/s41598-022-19998-5; https://doi.org/10.1093/biostatistics/kxad029). Clarifying the benefits of the GA relative to these methods would strengthen the methodological section.
3. *Choice of DBSCAN for clustering drug cocktails.*
I suggest elaborating on why DBSCAN was selected to cluster similar drug cocktails. What representation of drug combinations was used for clustering, and what interpretability benefits does DBSCAN bring?
- *FAERS data processing and selection.*
1. Please clarify why only two years were included and why specifically 2013–2015 were selected. Was this choice tied to the case study, data quality, or computational constraints?
2. Regarding deduplication, was this done using only report ID fields (primaryid or caseid), or was the rule‑based deduplication approach (e.g., that implemented in the DiAna R package) considered?
3. I suggest acknowledging the limitations of the DiAna drug dictionary and its partial mapping to ATC. In particular, DiAna does not handle ATC combination codes but instead splits combination products into their individual active substances (see https://doi.org/10.1007/s40264-023-01391-4).
- *Clarifications on Figure 2.*
It is unclear why the PRR is represented as a single point. Skimming the cited reference (https://doi.org/10.1002/sim.8457), the PRR does not look like a binary measure. I suggest double‑checking the computation and representation. If the PRR is shown as a single point, should the RR also appear as such? Clarification would help avoid confusion.
- *Simulation outputs for RR and PRR.*
In Figure 2, the x‑axis distributions for both RR and PRR span up to ~3000 and appear very similar. I recommend reviewing the simulation code to ensure these distributions are behaving as expected.
- *Omega computations.*
It is unusual for the Omega shrinkage metric not to exceed zero in any instance. I suggest verifying the implementation and/or simulation parameters. You may also compare with the values derived from FAERS data to determine whether the behaviour persists in real data.

Minor suggestions

- While adopting ATC as the drug hierarchy is appropriate, it may be worth briefly discussing whether the method could be applied to other drug classification systems, especially in therapeutic areas where ATC granularity may be limited.
- The reference to Zhang et al. (2015) currently appears in the Results. I suggest introducing this work earlier, ideally in the Introduction, when summarising prior research on the topic.

**Strengths And Weaknesses:**

Strengths
- Relevant and timely research question: The paper addresses the important problem of identifying potential drug–drug interactions (DDIs) from spontaneous reporting systems (SRS), a topic of high relevance for pharmacovigilance and public health.
- Novel computational perspective: The authors propose a computational framework that treats DDIs as multi‑drug “cocktail” hazards and explores their risk profiles. This conceptual shift is innovative compared to classical pairwise DDI analyses.

Weaknesses
- Insufficient justification of methodological choices: Several key modelling decisions (hypergeometric distribution, genetic algorithm search, DBSCAN clustering) are not fully justified, especially given that these choices are uncommon in pharmacovigilance. Established alternatives (subset scanning methods, tree‑based scan statistics, disproportionality‑based approaches) are not always discussed as comparators.
- Computational feasibility not explored: Although the proposed method is evaluated on a relatively small FAERS subset, it remains unclear whether the approach is computationally feasible at full real-world scale. Spontaneous reporting systems such as FAERS or VigiBase contain millions of reports, and the number of possible drug combinations increases combinatorially with the number of co-reported substances. It would be important for the authors to discuss whether the method remains tractable when applied to the complete database. Some estimation of computational complexity, runtime, memory requirements, or sub‑sampling strategy would greatly improve clarity on the method’s scalability.
- Limitation to a single case study: While expanding the scope may not be the primary aim of this article, applying the method to full datasets and additional case studies would better demonstrate its clinical utility. In addition, involving clinical experts in the interpretation of large drug‑cocktail signals would enhance the reliability and usefulness of the results.

---

> ### Author Response · Authors · 2026-01-16
> **First answers to specific remarks of reviewer YBby**
>
> ## Weaknesses
>
> **Insufficient justification of methodological choices: Several key modelling decisions (hypergeometric distribution, genetic algorithm search, DBSCAN clustering) are not fully justified, especially given that these choices are uncommon in pharmacovigilance. Established alternatives (subset scanning methods, tree‑based scan statistics, disproportionality‑based approaches) are not always discussed as comparators.**
>
> We investigate this point in further details in the next subsection, Major suggestions.
>
> **Computational feasibility not explored: Although the proposed method is evaluated on a relatively small FAERS subset, it remains unclear whether the approach is computationally feasible at full real-world scale. Spontaneous reporting systems such as FAERS or VigiBase contain millions of reports, and the number of possible drug combinations increases combinatorially with the number of co-reported substances. It would be important for the authors to discuss whether the method remains tractable when applied to the complete database. Some estimation of computational complexity, runtime, memory requirements, or sub‑sampling strategy would greatly improve clarity on the method’s scalability.**
>
>
> We agree that the combinatorial nature of drug interactions is a major hurdle in pharmacovigilance. We have updated the manuscript (see Appendix D) to explicitly address the method’s complexity and its tractability at a real-world scale.
>
>
> The primary advantage of our framework is that it decouples the search process from the combinatorial explosion of all possible drug combinations. By using a Genetic Algorithm  and Markov Chain Monte Carlo approach, the algorithm evaluates a fixed number of cocktails (determined by population size and number of iterations) rather than performing an exhaustive search.
>
> As detailed in the new Algorithmic Complexity section, the evaluation of a single drug cocktail scales linearly with the number of reports ($N$). The number of cocktails beeing evaluated has to be chosen by the user through the number of runs and iterations, finding a cut-off between the execution time and the exploration of the space of possibilities.
>
>
> **Limitation to a single case study: While expanding the scope may not be the primary aim of this article, applying the method to full datasets and additional case studies would better demonstrate its clinical utility. In addition, involving clinical experts in the interpretation of large drug‑cocktail signals would enhance the reliability and usefulness of the results.**
>
>
> As mentioned in the preamble, applications to novel data with real pharmacological implications are planned on hospital data with medical collaborators. They however will need more time for data collection and result analysis, and should be published in medical journals, less focused on methodological developments.
>
> Furthermore, one of the authors (K. Svendsen) is a pharmacist. He notably annotated by hand the myopathy results, without knowing the found clusters, which allowed to illustrate the coherence of the results.
>
>
> ## Major suggestions (Part 1)
>
> ***Limitations of spontaneous reporting systems (SRS)*. I recommend adding a discussion on the inherent limitations of SRS data regarding drug reporting. Due to the spontaneity of reporting, not all drugs taken are necessarily recorded, and the timing of exposures is not always specified. As a result, reported drugs may reflect simultaneous intake or different time windows, potentially limiting the accuracy of interaction analyses. Additionally, although FAERS allows reporters to specify interacting agents, this field is rarely populated. It may be informative to report how many cases in your FAERS subset include interacting substances.**
>
> We have expanded the Introduction to explicitly discuss the "snapshot" nature of SRS and the associated limitations regarding temporal precision and reporting completeness.
>
> Regarding the "interacting agents" field in FAERS, we deliberately chose to evaluate all co-reported medications for each patient rather than relying on this specific field. In our FAERS subset of 1,612,931 patients, we found that this field is sparsely and inconsistently populated. To quantify this, we searched each report for two indicators:
>
> - A drug "role" (characterization) coded as 3, indicating a suspected interaction.
>
> - The presence of the word "interaction" in the MedDRA Preferred Terms associated with the reaction.
>
>
> In total, only $18,014$ reports (approximately $1.1\%$ of the dataset) were identified using these criteria, in accordance with the work of Alahmari et al. (2025). Due to the low frequency and poor reliability of that information,  we decided not to take it into account.

---

> ### Author Response · Authors · 2026-01-16
> **Second answers to specific remarks of reviewer YBby**
>
> ## Major suggestions (Part 2)
>
> ***Choice of the hypergeometric distribution.* Please provide a more detailed rationale for modeling the hazard of drug combinations using a hypergeometric distribution. Were alternative distributions considered? Why was the hypergeometric deemed most appropriate? Is there precedent in the literature, or was this decision model‑driven by the authors? Although rows 99–102 compare your measure to the PRR, other methods commonly applied to drug–drug interactions (DDIs) may offer relevant contrasts. For instance, the Omega method is based on a Gamma–Poisson model. What advantages does the hypergeometric approach provide relative to these well‑established alternatives?**
>
> The hypergeometric p-value corresponds, as pointed by reviewer 7TJo, to the p-values of the Fisher test of the contingency table of AE occurrence vs cocktail consumption. Being a model-independent disproportionality analysis measure, it has the advantage to be easily generalized to cocktails of any size. This is crucial for our approach and not possible for instance using the Omega method. It was moreover used in Ahmed (2010), and the citation was added as suggested by 7TJo.  We rewrote Section 2.2 to include more details about the compared scoring functions.
>
> ***Choice of a genetic algorithm (GA) as a search method.* To my knowledge, genetic algorithms are rarely used in pharmacovigilance, and more broadly, heuristic search algorithms have not been widely applied in spontaneous reporting systems. I strongly recommend adding a justification for preferring a GA over established SRS search or screening approaches, such as subset‑based methods [...] or tree‑based scan statistics [...]. Clarifying the benefits of the GA relative to these methods would strengthen the methodological section.**
>
> As underlined in the preamble, the main objective and novelty of our approach is to explore the set of all drug cocktails looking for associations with a specific AE. A drug cocktail is a set of nodes of the ATC tree, each node representing either a single drug if it is a leaf or a set of drugs if it is an internal node (equivalent to a cut in the tree-scan approach). Exploring all drug cocktails means that:
>
> - they can involve any number of nodes
> - their nodes can be spread in any way in the tree
>
> The space to explore therefore formally contains $2^L$ elements, where $L$ is the number of nodes of the tree, which is of course way too large for an exhaustive search. This is the reason for which we rely on GA, which allows a random exploration  driven towards interesting regions of the huge space of possible solutions.
>
> We have the impression that the usual approaches, including the suggested methods, do not answer the same question. The subset-based method allows to use a refined score based on stratifications of confounding variables, but for the association of a single drug and a given AE. This score could probably be generalized to cocktails. However, the screening of all possible associations is done via a *For* loop trying them all, which is what has to be avoided for the problem we consider.
>
> The tree-based scan statistic looks for interesting cuts in the ATC tree, thus only grouping the drugs that are in a common subtree. Disproportionality analysis is run by comparing the proportion of patient experiencing the AE among those taking one of those drugs with the proportion among the patients taking none. In our method, the hypergeometric score of a cocktail of size $1$ corresponds to the disproportionality in the subtree containing the descendant of the considered node. Even if technically different, finding the best cut with tree-scan is comparable to finding the best cocktail of size $1$ in our method. It can therefore not be compared to the GA algorithm which has a different purpose by considering cocktails of several drugs or family of drugs taken simultaneously and potentially far away from each other in the ATC tree.
>
>
> We added in the introduction a reference to the tree-based scan statistic and the fact that it takes the tree-structure into account, and modified the method section to insist on the specificity of our problem and the need of a non-exhaustive search.

---

> ### Author Response · Authors · 2026-01-16
> **Third answers to specific remarks of reviewer YBby**
>
> ## Major suggestions (Part 3)
> ***Choice of DBSCAN for clustering drug cocktails.* I suggest elaborating on why DBSCAN was selected to cluster similar drug cocktails. What representation of drug combinations was used for clustering, and what interpretability benefits does DBSCAN bring?**
>
> DBSCAN was selected because it is a density-based algorithm that does not require a pre-defined number of clusters and is highly effective at identifying clusters of arbitrary shapes while handling noise.
>
> The main advantage of the DBSCAN algorithm is that it is representation-free: it can perform clustering by taking only the similarity matrix between objects to be clustered as input. In our case, it is thus sufficient to define a custom editing distance, adapted to treat cocktails as unordered sets within the ATC hierarchy.
>
> As the similarity between cocktails is based on an interpretable tree-based metric, it leads to clear interpretability benefits: cocktails within the same cluster should exhibit pharmacological similarities, allowing clinicians to interpret the results as a few distinct signals rather than hundreds of individual, redundant rows. Ultimately, we believe that the construction of a meaningful, tree-aware distance metric is more critical to obtaining high-quality clusters than the specific choice of the clustering algorithm itself.
>
> ***FAERS data processing and selection.* Please clarify why only two years were included and why specifically 2013–2015 were selected. Was this choice tied to the case study, data quality, or computational constraints?**
>
> The selection of the two-year window from Q2 2013 to Q2 2015 was guided by two primary factors:
>
> - **Data Standardization:** We chose to begin the study following the major transition in FAERS data architecture that occurred in September 2012 (transition from AERS to FAERS). Selecting reports from 2013 onward ensured that the data structure was more consistent, facilitating the extraction and active ingredient mapping.
>
> - **Iterative Methodology Development:** As this study introduces a novel computational framework involving a Genetic Algorithm and MCMC sampling, the development process required numerous iterations to ensure the correctness of the method. Restricting the initial proof-of-concept to a two-year window provided a practical balance between sample size and computational runtime, allowing for the rapid experimentation necessary to refine the algorithms. Despite the restricted time window, the resulting dataset included over 1.6 million patients
>
> While our previous response demonstrates that the method is linearly scalable and capable of handling larger datasets, we found this two-year period to be an optimal real-world subset for validating the methodology's signal detection capabilities.
>
> **Regarding deduplication, was this done using only report ID fields (primaryid or caseid), or was the rule‑based deduplication approach (e.g., that implemented in the DiAna R package) considered?**
>
> In this study, we performed deduplication by identifying reports with the same case identifier and retaining only the version with the most recent report ID.
>
> We did not implement the rule-based deduplication on demographics similarity found in packages like DiAna for this specific study. We used the DiAna dictionary specifically for the standardization of drug names to ATC codes.
>
> **I suggest acknowledging the limitations of the DiAna drug dictionary and its partial mapping to ATC. In particular, DiAna does not handle ATC combination codes but instead splits combination products into their individual active substances (see https://doi.org/10.1007/s40264-023-01391-4).**
>
> We have updated Section 2.6.2 to acknowledge that DiAna decomposes combination products into their individual active ingredients rather than using specific ATC combination codes.
>
> In the context of our methodology, this behavior could be considered beneficial. Since our framework aims to identify interactions among sets of active substances, treating combination products as a cocktail of individual ingredients allows the algorithms to evaluate the risk associated with each specific component. This ensures that a signal driven by a single substance within a combination product can still be identified. We agree that this approach may lose some information regarding the specific formulation of combination products (e.g. posology), but it can enhances the algorithm in a way.

---

> ### Author Response · Authors · 2026-01-16
> **Fourth and final answers to specific remarks of reviewer YBby**
>
> ## Major suggestions (Part 4)
>
> ***Clarifications on Figure 2.* It is unclear why the PRR is represented as a single point. Skimming the cited reference (https://doi.org/10.1002/sim.8457), the PRR does not look like a binary measure. I suggest double‑checking the computation and representation. If the PRR is shown as a single point, should the RR also appear as such? Clarification would help avoid confusion.**
>
> The single point in Figure 2 represents the PRR adaptation for DDI as defined by Wang et al. (2020), rather than the standard continuous PRR calculation for single drugs described by Evans et al. (2001). While the standard PRR is indeed a continuous measure, the DDI-specific application follows a binary signaling rule to identify synergistic risks. As stated in the end of their Section 2, a DDI signal is defined by a specific condition: \"Similar to the approach proposed by Almenoff et al, a drug pair with $PRR_{025 D1D2} > \max(PRR_{025 D1}, PRR_{025 D2})$ is considered as a signal of DDI\".
>
> **Simulation outputs for RR and PRR. In Figure 2, the x‑axis distributions for both RR and PRR span up to ~3000 and appear very similar. I recommend reviewing the simulation code to ensure these distributions are behaving as expected.**
>
> After reviewing our plotting scripts, we confirm that the jitter plot labeled "PRR" displayed the values for the Relative Risk (RR), because the PRR (as defined by Evans et al., 2001) and the RR are computed using similar ratios from the contingency table. This should be a single plot, which was modified accordingly.
>
> **Omega computations. It is unusual for the Omega shrinkage metric not to exceed zero in any instance. I suggest verifying the implementation and/or simulation parameters. You may also compare with the values derived from FAERS data to determine whether the behaviour persists in real data.**
>
> We carefully reviewed the computation of the Omega shrinkage. After review, it seems that there are no errors, we compared our results with another implementation that can be found here (https://rpubs.com/WilluOttawa/ddiomega) that is directly derived from the review from Noguchi et al. (2019). There is an agreements between computed values with both method.
>
> We tried to compute the Omega shrinkage on size two cocktails that are on the top of our final results list on the FAERS dataset. On the colchicine/cyclosporin cocktail, the Omega Shrinkage yields a value of $3.08$ declaring it as a signal as well as our method.
>
> ## Minor suggestions
>
> **While adopting ATC as the drug hierarchy is appropriate, it may be worth briefly discussing whether the method could be applied to other drug classification systems, especially in therapeutic areas where ATC granularity may be limited.**
>
> As long as the drug classification system can be represented as a tree, the method could be applied. The conclusion briefly discusses the fact that this method could be applied to other non drug hierarchy systems like ICD or MedDRA.
>
> **The reference to Zhang et al. (2015) currently appears in the Results. I suggest introducing this work earlier, ideally in the Introduction, when summarizing prior research on the topic.**
>
> This reference now appears in the introduction.
>
> ## Broader impact concerns
>
> **Although the paper does not include a Broader Impact Statement, I believe adding one would be appropriate given the public‑health implications of the work. This study develops an algorithmic framework to identify potential drug–drug interactions using spontaneous reporting systems (SRS). Such methods, if misinterpreted or deployed without appropriate safeguards, may influence clinical decision-making, regulatory considerations, or public perception of drug safety.**
>
> **I would therefore encourage the authors to briefly reflect on:**
>
> - **The risk of misinterpretation of algorithmically identified associations, especially since SRS data suffer from underreporting, missing temporal information, and reporting biases.**
>
> - **The possibility of false positives or false negatives, and how these may affect patient safety if results are communicated without appropriate caveats.**
>
> - **The ethical responsibility to frame outputs as hypothesis‑generating rather than causal, particularly when using heuristic search algorithms (GA, DBSCAN) whose outputs are not statistically calibrated.**
>
> - **The potential impact on stakeholders (clinicians, regulators, patients) if high‑risk drug combinations identified through the method are disseminated without appropriate contextualization.**
>
> **A short paragraph addressing these points would strengthen the responsible‑research framing of the work.**
>
>
> A broader impact statement has been added at the very end of the conclusion.

---

### Comment · Editors_In_Chief · 2025-10-02
**Access to your submitted manuscript prior to reviewing**

Dear authors,

Thanks for your submission 👍

Could you please add a link to the deployed gh-page in your repository description/about, to facilitate the reviewing process?
https://bangard.xyz/202509-bangard-detecting/

By the way, for your information: the labeling problem in the algorithm blocks was fixed in the pseudocode extension, and we integrated this in the last version of https://github.com/computorg/computo-quarto-extension, see https://github.com/computorg/computo-quarto-extension/issues/43. You can safely label your algorithm using the prefix `algo-`.

Best,

---

### Comment · Action_Editor_dfKG · 2025-10-07
**Required submission updates for reproducibility**

Dear authors,

The git repository of your submission seem to contain all the necessary information in https://github.com/JulesBa-Git/202509-bangard-detecting/tree/main/figures/scripts. However, putting all the code in a dedicated subfolder instead of the notebook results in a separation between the code and the main text of the paper, which can be an obstacle to reproducibility.

Therefore we kindly ask you to include in your .qmd notebook the code to generate all the figures. We do not require all the computations to be done at each compilation: for long computations you can store pre-computed results, as you already did for some of your code.

We will then be able to send your paper for review.

Do not hesitate to reach out if you need clarifications.

---

> ### Author Response · Authors · 2025-10-09
> **Reproducibility updates completed**
>
> Dear Action Editor,
>
> Thank you for your feedback. We have updated our repository accordingly, each figure can now be generated from the corresponding code chunks within the ```.qmd``` notebook, as requested.
>
> Please let us know if any further modifications are needed.
>
> Best regards,

---

> > ### Comment · Action_Editor_dfKG · 2025-10-14
> > **need to set up renv**
> >
> > Thanks for addressing this point quickly.
> >
> > One final point before we send your contribution for review: can you setup renv as described in our [guidelines](https://computo-journal.org/site/guidelines-authors.html#sec-dependencies)? This will make it easier to reproduce your work.

---

> > > ### Comment · Action_Editor_dfKG · 2025-11-14
> > > **Paper sent for review**
> > >
> > > The paper has been sent for review.

---

### Comment · Action_Editor_dfKG · 2026-01-05
**Start of rebuttal period**

Dear authors,

We have received two reports for your submission to Computo.

A period of 6 weeks (starting today) is allowed for discussion with the referees before they issue a final opinion. During this period, you can make any changes to your submission that you feel are necessary and that you are able to make. If you need more time for this rebuttal, please let us know. At the end of this period, a decision will be made, ranging from final acceptance to rejection. Please note further major revisions are handled via a resubmission.

Best regards

---

### Author Response · Authors · 2026-01-16
**General answer to reviews**

We first would like to thank to the reviewers for the careful reading and insightful comments, which have allowed us to significantly improve the first version of our manuscript.

Regarding the general remarks, we would like to emphasize that the goal of the method is to list the sets of medications (including single drugs) that significantly increase the risk of developing a given Adverse Event(AE). Its novelty lies in its ability to account for:

- drug cocktails of any size
- the hierarchical structure of the ATC classification, which enables  the detection of interactions involving entire drug families

The scientific challenge addressed — and which we believe aligns with the scope of Computo— is not a specific application but rather the development of an innovative methodology to explore the space of possible drug cocktails with those objectives in mind and in a computationnally efficient way. The primary contributions of this work are therefore the proposal of the genetic and MCMC algorithms based on the ATC tree. The simulated and real-world applications should be viewed as proofs of concept rather than for their applied findings. Applications to novel data with real pharmacological implications are planned on hospital data with medical collaborators. They however will need more time for data collection and result analysis, and should be published in medical journals, less focused on methodological developments. We think that Computo is the right journal to provide the methodological descriptions and discuss illustrations on simulated and real datasets.

In light of these considerations, we have substantially revised several key sections of the manuscript:

- We were previously unaware that the hypergeometric score had already been employed in the form of Fisher’s exact test. We have corrected this oversight and revised the section on score selection. Additionally, we clarified that the score is not fixed to our hypergeometric choice; the method can accommodate different scoring functions (some possibilities will be included in the next version of the package).

- We hope to have provided a clearer explanation of the algorithms. We have specified that their aim is to identify all sets (of size $1$, $2$, or more) for which $H(C)$ is significantly high. As a reviewer pointed out, for multiple drugs some of them may not correspond to interactions in the sense that the effect could resume to the sum of the single drug effects. Any post-processing step of the literature may be used to decide if the selected cocktails' effect are due to a proper interaction. The possibility to do it through a logistic regression including interaction terms has been added  to the package. The addition of more sophisticated methods will be considered in future package development.

- We have redefined the notion of the p-value provided by the MCMC algorithm. As a reviewer correctly noted, it corresponds to the survival function value of $H(C)$ under its distribution — a mixture of $H_0$ (cocktails without effect) and $H_1$ (cocktails promoting the adverse event, AE). Given that only a small fraction of cocktails promote an AE within the vast space of possible cocktails, we considered this a reasonable approximation of the p-value. We still think it is except in the tail of the distribution, where cocktails have a high enough score to justify an expertise by a pharmacologist. It was however not precise enough, and we have discussed this point accordingly.

- We hope that the revised presentation of the real-world dataset application, framed as a proof of concept, is now clearer.

We hope that those modifications answer the general comments about the paper. Answer to the specific comments of the reviewers are further discussed point per point in response to each review.

---

### Comment · Action_Editor_dfKG · 2026-02-02
**End of rebuttal period: reviewers should now submit their official recommendation**

Dear reviewers,

The authors have responded to your comments (see below) and produced an updated version of their manuscript, available here :

  https://bangard.xyz/202509-bangard-detecting/

You should now, within the next two weeks, make a recommendation based on the authors' responses and the updated version of the submission. To do this, please use the "Official Recommendation" box. On the basis of your recommendations, the editorial board will make a decision (acceptance, major or minor revisions, rejection).

Thank you for your time in reviewing this manuscript.

---

### Comment · Reviewer_7TJo · 2026-02-04

Thank you for carefully addressing my comments.

-	I still have a concern about the positioning of the paper (in the title)
The authors have added some ad hoc analyses or sentences to better present their method as a method to detect dug interaction. In particular, they added a paragraph indicating that logistic regression, including an interaction term, can be applied to the clusters identified by their method for size-2 drug interactions. They also included a discussion of their findings regarding the application to FAERS data in the conclusion section. However, in my opinion, detecting interactions is not central to the proposed method which is about detecting drug cockails over-represented in the data. It was not central in the initial submission, nor is it central in the revised version despite the additions. For example, the simulation does not demonstrate the ability to detect interactions. Discussion of identifying potential true interactions in the application is deferred to the conclusion of the paper, not the results. Logistic regression is only quickly mentionned for detecting size-2 interactions, but the title mentions high-order interactions.
Therefore, my suggestion would be to modify the manuscript to present the method primarily as a way to detect overrepresented drug cocktails (starting from the title and abstract).

-	I have another concern regarding the simulation study.
Regarding the simulation study, it is still unclear to me which data were used for which analysis. In Section 2.6.1, the authors mention creating four datasets. It would be helpful to number them and refer to these numbers in the results section (and the corresponding figures).
Secondly, in Section 3.2.2 the authors refer to the similarity of clusters based on pharmacological families (« This post-processing step allows to reduce redundancy by grouping cocktails that differed only slightly, such as by substituting a drug for another within the same pharmacological family »).  This makes me realize that I am missing some information to understand how the data were simulated in Section 2.6.1. For instance, how many drugs were considered in total? How was the ATC structure used and reproduced? How were C1, C2, C3, and C4 chosen in the ATC hierarchy? Could they be chosen at different levels of the hierarchy? If so, that would be somewhat strange, since, despite the potential existence of a pharmacological effect (meaning that several drugs in the same class have the same AE), people take drugs (ATC5), not drug classes.
I would thus suggest that the authors describe their simulation procedure more precisely.

---

### Author Response · Authors · 2026-04-04
**Minor revision answer to reviewer and action editor**

We thank the reviewers and the action editor for their careful reading and constructive feedback. We addressed each point.

---

## 1. Positioning of the paper (title and framing)

**Reviewer comment:** *The reviewer notes that detecting interactions is not central to the proposed method, which is primarily about detecting drug cocktails overrepresented in the data, and suggests modifying the manuscript accordingly — starting from the title and abstract.*

**Response:**

We agree with the reviewer's assessment. The core contribution of our method is indeed the detection of drug cocktails that are overrepresented in adverse event reports. The interaction analysis via penalized logistic regression is a post-treatment step that complements the main pipeline but is not its central purpose.

We have accordingly revised the manuscript as follows:

- The title has been changed to: *"Detecting adverse high-order drug combinations from individual case safety reports using computational statistics on disproportionality measures"*.
- The abstract has been revised to emphasize that the method identifies overrepresented drug cocktails, and to present the logistic regression step as an optional post-treatment for distinguishing true interactions from combined individual effects, rather than as a core feature.
- Throughout the manuscript, we have carefully revised phrasing to ensure that the method is consistently presented as a signal detection tool for overrepresented cocktails, with interaction assessment positioned as a downstream analysis step.

We believe this revised framing more accurately reflects the method's contributions while still acknowledging the practical relevance of the interaction assessment for end users.

---

## 2. Simulation study: clarity and data description

**Reviewer comment:** *The reviewer requests clearer numbering of the four simulated datasets with consistent references in the results, and asks for more details on how the ATC structure was used — in particular, how C1–C4 were chosen in the hierarchy and whether patients are assigned drugs at the leaf level.*

**Response:**

We thank the reviewer for pressing on this point, which led to improvements in the clarity of the simulation section.

We have made the following changes:

- The four datasets are now explicitly numbered (D1, D2, D3, D4) in Section 2.6.1, and all references in the results section and figure captions now use these identifiers consistently.
- We have added a detailed description of how the simulation uses the ATC tree.

We hope these additions address the reviewer's concern. We are happy to provide further details if needed.

---

## 3. Typos noted by the Action Editor

We thank the action editor for spotting these typos, all have been corrected. We have also performed a thorough proofreading pass on the entire manuscript.

We thank the reviewer and the action editor once more for their guidance throughout this process. We believe the revised manuscript now provides a clearer and more accurate presentation of the method's scope and contributions.

---

> ### Comment · Action_Editor_dfKG · 2026-04-13
> **Acceptance and reproducibility checks**
>
> Dear authors,
>
> Thank you very much for these modifications which address the remaining concerns raised.
>
> Before we can proceed to the editorial production of your article, your contribution needs to pass the required reproducibility criteria for Computo. This is a follow up to a previous comment before review: https://openreview.net/forum?id=Bcm0PqmKkq&noteId=fqyizc4QYO
>
> Indeed, your github repository contains code to reproduce the results shown in the paper, the code in your .qmd notebook is only generating the figures from binary data files. We acknowledge that the computations may be too long to be run at each compilation. However, we require that your .qmd notebook includes an illustration of usage of your method in a small example, see https://computo-journal.org/blog/2023-06-21-long-running-code/
>
> Perhaps this could be done either for (part of) Dataset D1 or the FAERS data? If not, could you add a small subsection in Appendix showcasing the use of the method on one of the datasets already available in the emcAdr package?

---

> > ### Author Response · Authors · 2026-05-01
> > **Addition of an illustrative example**
> >
> > Dear Editor,
> >
> > We have addressed the request by adding Appendix E: Illustrative Example of Package Usage to the manuscript. The appendix walks through the emcAdr pipeline end-to-end on a small simulated dataset (5,000 patients, with two solutions). It compiles in a few minute as part of the .qmd notebook. It covers: dataset generation, computation of the hypergeometric score $H$; MCMC approximation of the null distribution and the corresponding empirical p-value; recovery of the solutions via the genetic algorithm; UMAP+DBSCAN clustering of the recovered cocktails and the Firth penalized logistic regression post-treatment.
> >
> > The simulation design mirrors that of Dataset $D_2$ described in Section 2.6.1, with adverse-event probabilities deliberately inflated so that the signals remain detectable at the reduced sample size. The updated manuscript and rendered notebook have been pushed to the article's GitHub repository.

---

### Comment · Action_Editor_dfKG · 2026-05-13
**Production of your article**

Dear authors,

Thank you for adding the illustrative example as requested. Your paper is now formally accepted and we are ready to enter the production step.

Can you double check the metadata of the article, ie the fields `affiliation`, `author-url`, `affiliation-url` are correctly filled in https://github.com/JulesBa-Git/202509-bangard-detecting/blob/main/_quarto.yml ?

I have just invited the corresponding author (JulesBa-Git) as a collaborator of computorg, the github organization of Computo. Please transfer the ownership of your repository to computorg as follows:  at the bottom of https://github.com/JulesBa-Git/202509-bangard-detecting/settings, click  "Transfer ownership" and choose "computorg" in "Select one of my organizations".

Then we will be able to proceed with the publication of your article.

Thanks again for submitting your work to Computo!

---

### Note · Reviewer_7TJo · 2026-02-04

**Comment:**

I have two last comments (in line with previous ones) that I think are important.

One concerns the positioning of the paper. The authors present their method as one for detecting high-order drug interactions, starting from the title. However, "interaction" has a precise meaning, especially in pharmacoepidemiology and pharmacovigilance. Despite the addition of some elements following my first comment, I still think the method should not be presented that way or at least, I don’t think that the authors evaluated the method in its ability of detecting high order interactions.
The other comment pertains to the simulation study, which I still find unclear, particularly in terms of how the data are simulated and how the results are presented.
(So for me, the answer to the claims and evidence field is not a clear yes)

I'll let you judge the importance of these two final comments as I will not have much time to dedicate to this work.

Best regards,
Ismaïl Ahmed

**Audience:**

Yes

**Claims And Evidence:**

Yes

**Decision Recommendation:**

Leaning Accept

---

### Note · Reviewer_YBby · 2026-02-20

**Comment:**

I agree with the comments from the other reviewer and suggest that they be addressed before final acceptance. I am still not fully clear on how the simulations were performed (also related to the Omega performance—see my previous revision).

**Audience:**

Yes

**Claims And Evidence:**

Yes

**Decision Recommendation:**

Leaning Accept

---

### Decision · Action_Editor_dfKG · 2026-03-16

**Recommendation:** Accept with minor revision

**Comment:**

Both reviewers have noted that the authors have addressed most of the points raised during the first round of decision.

They have also noted that two points should be further clarified, regarding the positioning of the paper and the simulation study, see the detailed comment of Reviewer 7TJo on this point.

Note: I have also spotted some typos (please proofread your paper carefully):

- line 75: inconsistent author naming in citation
- line 96: missing space in "cocktails.The"
- line 299: "This enable"

**Audience:**

Fits well with the scope of Computo

**Claims And Evidence:**

In agreement with the elements provided by the reviewers, I consider that the claims and evidence in this submission are now sufficiently solid for publication, provided that two points regarding the positioning of the paper and the simulation study are clarified.

---

> ### Decision · Editors_In_Chief · 2026-03-23
>
> I approve the AE's decision.